# Average Case Column Subset Selection for Entrywise $\ell_1$-Norm Loss

**Zhao Song**[*]
University of Washington
magic.linuxkde@gmail.com

**David P. Woodruff**[*]
Carnegie Mellon University
dwoodruf@cs.cmu.edu

**Peilin Zhong**[*]
Columbia University
pz2225@columbia.edu

## Abstract

We study the column subset selection problem with respect to the entrywise $\ell_1$-norm loss. It is known that in the worst case, to obtain a good rank-$k$ approximation to a matrix, one needs an arbitrarily large $n^{\Omega(1)}$ number of columns to obtain a $(1 + \epsilon)$-approximation to the best entrywise $\ell_1$-norm low rank approximation of an $n \times n$ matrix. Nevertheless, we show that under certain minimal and realistic distributional settings, it is possible to obtain a $(1+\epsilon)$-approximation with a nearly linear running time and $\text{poly}(k/\epsilon) + O(k \log n)$ columns. Namely, we show that if the input matrix $A$ has the form $A = B + E$, where $B$ is an arbitrary rank-$k$ matrix, and $E$ is a matrix with i.i.d. entries drawn from any distribution $\mu$ for which the $(1 + \gamma)$-th moment exists, for an arbitrarily small constant $\gamma > 0$, then it is possible to obtain a $(1 + \epsilon)$-approximate column subset selection to the entrywise $\ell_1$-norm in nearly linear time. Conversely we show that if the first moment does not exist, then it is not possible to obtain a $(1 + \epsilon)$-approximate subset selection algorithm even if one chooses any $n^{o(1)}$ columns. This is the first algorithm of any kind for achieving a $(1 + \epsilon)$-approximation for entrywise $\ell_1$-norm loss low rank approximation.

## 1 Introduction

Numerical linear algebra algorithms are fundamental building blocks in many machine learning and data mining tasks. A well-studied problem is low rank matrix approximation. The most common version of the problem is also known as Principal Component Analysis (PCA), in which the goal is to find a low rank matrix to approximate a given matrix such that the Frobenius norm of the error is minimized. The optimal solution of this objective can be obtained via the singular value decomposition (SVD). Hence, the problem can be solved in polynomial time. If approximate solutions are allowed, then the running time can be made almost linear in the number of non-zero entries of the given matrix [1, 2, 3, 4, 5, 6].

An important variant of the PCA problem is the entrywise $\ell_1$-norm low rank matrix approximation problem. In this problem, instead of minimizing the Frobenius norm of the error, we seek to minimize the $\ell_1$-norm of the error. In particular, given an $n \times n$ input matrix $A$, and a rank parameter $k$, we want to find a matrix $B$ with rank at most $k$ such that $\|A - B\|_1$ is minimized, where for a matrix $C$, $\|C\|_1$ is defined to be $\sum_{i,j} |C_{i,j}|$. There are several reasons for using the $\ell_1$-norm as the error measure. For example, solutions with respect to the $\ell_1$-norm loss are usually more robust than solutions with Frobenius norm loss [7, 8]. Further, the $\ell_1$-norm loss is often used as a relaxation of the $\ell_0$-loss, which has wide applications including sparse recovery, matrix completion, and robust PCA; see e.g., [9, 8]. Although a number of algorithms have been proposed for the $\ell_1$-norm loss [10, 11, 12, 13, 14, 15, 16, 17, 18, 19, 20, 21, 22], the problem is known to be NP-hard [23]. The

---

[*]equal contribution.

first $\ell_1$-low rank approximation with provable guarantees was proposed by [24]. To cope with NP-hardness, the authors gave a solution with a $\mathrm{poly}(k \log n)$-approximation ratio, i.e., their algorithm outputs a rank-$k$ matrix $B' \in \mathbb{R}^{n \times n}$ for which

$$\|A - B'\|_1 \le \alpha \cdot \min_{\mathrm{rank}-k\ B} \|A - B\|_1 \tag{1}$$

for $\alpha = \mathrm{poly}(k \log n)$. The approximation ratio $\alpha$ was further improved to $O(k \log k)$ by allowing $B'$ to have a slightly larger $k' = O(k \log n)$ rank [25]. Such $B'$ with larger rank is referred to as a bicriteria solution. However, in high precision applications, such approximation factors are too large. A natural question is if one can compute a $(1 + \epsilon)$-approximate solution efficiently for $\ell_1$-norm low rank approximation. In fact, a $(1 + \epsilon)$-approximation algorithm was given in [26], but the running time of their algorithm is a prohibitive $n^{\mathrm{poly}(k/\epsilon)}$. Unfortunately, [26] shows in the worst case that a $2^{k^{\Omega(1)}}$ running time is necessary for any constant approximation given a standard conjecture in complexity theory.

**Notation.** To describe our results, let us first introduce some notation. We will use $[n]$ to denote the set $\{1, 2, \cdots, n\}$. We use $A_i$ to denote the $i^{\mathrm{th}}$ column of $A$. We use $A^j$ to denote the $j^{\mathrm{th}}$ row of $A$. Let $Q \subseteq [n]$. We use $A_Q$ to denote the matrix which is comprised of the columns of $A$ with column indices in $Q$. Similarly, we use $A^Q$ to denote the matrix which is comprised of the rows of $A$ with row indices in $Q$. We use $\binom{[n]}{t}$ to denote the set of all the size-$t$ subsets of $[n]$. Let $\|A\|_F$ denote the Frobenius norm of a matrix $A$, i.e., $\|A\|_F$ is the square root of the sum of squares of all the entries in $A$. For $1 \le p < 2$, we use $\|A\|_p$ to denote the entry-wise $\ell_p$-norm of a matrix $A$, i.e., $\|A\|_p$ is the $p$-th root of the sum of $p$-th powers of the absolute values of the entries of $A$. $\|A\|_1$ is an important special case of $\|A\|_p$, which corresponds to the sum of absolute values of the entries in $A$. A random variable $X$ has the Cauchy distribution if its probability density function is $f(z) = \frac{1}{\pi(1+z^2)}$.

## 1.1 Our Results

We propose an efficient bicriteria $(1 + \epsilon)$-approximate column subset selection algorithm for the $\ell_1$-norm. We bypass the running time lower bound mentioned above by making a mild assumption on the input data, and also show that our assumption is necessary in a certain sense.

Our main algorithmic result is described as follows.

**Theorem 1.1** (Informal version of Theorem 2.13). *Suppose we are given a matrix $A = A^* + \Delta \in \mathbb{R}^{n \times n}$, where $\mathrm{rank}(A^*) = k$ for $k = n^{o(1)}$, and $\Delta$ is a random matrix for which the $\Delta_{i,j}$ are i.i.d. symmetric random variables with $\mathbf{E}[|\Delta_{i,j}|^p] = O(\mathbf{E}[|\Delta_{i,j}|]^p)$ for some constant $p > 1$. Let $\epsilon \in (0, 1/2)$ satisfy $1/\epsilon = n^{o(1)}$. There is an $\widetilde{O}(n^2 + n \, \mathrm{poly}(k/\epsilon))^2$ time algorithm (Algorithm 1) which can output a subset $S \subseteq [n]$ with $|S| \le \mathrm{poly}(k/\epsilon) + O(k \log n)$ for which*

$$\min_{X \in \mathbb{R}^{|S| \times n}} \|A_S X - A\|_1 \le (1 + \epsilon)\|\Delta\|_1,$$

*holds with probability at least* $99/100$.

Note the running time in Theorem 1.1 is nearly linear in the number of non-zero entries of $A$, since for an $n \times n$ matrix with i.i.d. noise drawn from any continuous distribution, the number of non-zero entries of $A$ will be $n^2$ with probability 1. We also show the moment assumption of Theorem 1.1 is necessary in the following precise sense.

**Theorem 1.2** (Hardness, informal version of Theorem B.20). *Let $n > 0$ be sufficiently large. Let $A = \eta \cdot \mathbf{1} \cdot \mathbf{1}^\top + \Delta \in \mathbb{R}^{n \times n}$ be a random matrix where $\eta = n^{c_0}$ for some sufficiently large constant $c_0$, $\mathbf{1} \in \mathbb{R}^n$ is the all-ones vector, and $\forall i, j \in [n], \Delta_{i,j} \sim C(0, 1)$ are i.i.d. standard Cauchy random variables. Let $r = n^{o(1)}$. Then with probability at least $1 - O(1/\log \log n), \forall S \subseteq [n]$ with $|S| = r$,*

$$\min_{X \in \mathbb{R}^{r \times n}} \|A_S X - A\|_1 \ge 1.002\|\Delta\|_1.$$

## 1.2 Our Techniques

For an overview of our hardness result, we refer readers to the supplementary material, namely, Appendix B. In the following, we will outline the main techniques used in our algorithm.

$(1 + \epsilon)$**-Approximate $\ell_1$-Low Rank Approximation.** We make the following distributional assumption on the input matrix $A \in \mathbb{R}^{n \times n}$: namely, $A = A^* + \Delta$ where $A^*$ is an arbitrary rank-$k$ matrix and the entries of $\Delta$ are i.i.d. from any symmetric distribution with $\mathbf{E}[|\Delta_{i,j}|] = 1$ and $\mathbf{E}[|\Delta_{i,j}|^p] = O(1)$ for any real number $p$ strictly greater than 1, e.g., $p = 1.000001$ would suffice. Note that such an assumption is mild compared to typical noise models which require the noise be Gaussian or have bounded variance; in our case the random variables may even be heavy-tailed with infinite variance. In this setting we show it is possible to obtain a subset of $\text{poly}(k(\epsilon^{-1} + \log n))$ columns spanning a $(1 + \epsilon)$-approximation. This provably overcomes the column subset selection lower bound of [24] which shows for entrywise $\ell_1$-low rank approximation that there are matrices for which any subset of $\text{poly}(k)$ columns spans at best a $k^{\Omega(1)}$-approximation.

Consider the following algorithm: sample $\text{poly}(k/\epsilon)$ columns of $A$, and try to cover as many of the remaining columns as possible. Here, by covering a column $i$, we mean that if $A_I$ is the subset of columns sampled, then $\min_y \|A_I y - A_i\|_1 \le (1 + O(\epsilon))n$. The reason for this notion of covering is that we are able to show in Lemma 2.1 that in this noise model, $\|\Delta\|_1 \ge (1 - \epsilon)n^2$ w.h.p., and so if we could cover every column $i$, our overall cost would be $(1 + O(\epsilon))n^2$, which would give a $(1 + O(\epsilon))$-approximation to the overall cost.

We will not be able to cover all columns, unfortunately, with our initial sample of $\text{poly}(k/\epsilon)$ columns of $A$. Instead, though, we will show that we will be able to cover all but a set $T$ of $\epsilon n/(k \log k)$ of the columns. Fortunately, we show in Lemma 2.4 another property of the noise matrix $\Delta$ is that *all* subsets $S$ of columns of size at most $n/r$, for $r \ge (1/\gamma)^{1+1/(p-1)}$ satisfy $\sum_{j \in S} \|\Delta_j\|_1 = O(\gamma n^2)$. Thus, for the above set $T$ that we do not cover, we can apply this lemma to it with $\gamma = \epsilon/(k \log k)$, and then we know that $\sum_{j \in T} \|\Delta_j\|_1 = O(\epsilon n^2/(k \log k))$, which then enables us to run a previous $\widetilde{O}(k)$-approximate $\ell_1$ low rank approximation algorithm [25] on the set $T$, which will only incur total cost $O(\epsilon n^2)$, and since by Lemma 2.1 above the overall cost is at least $(1 - \epsilon)n^2$, we can still obtain a $(1 + O(\epsilon))$-approximation overall.

The main missing piece of the algorithm to describe is why we are able to cover all but a small fraction of the columns. One thing to note is that our noise distribution may not have a finite variance, and consequently, there *can* be very large entries $\Delta_{i,j}$ in some columns. In Lemma 2.3, we show the number of columns in $\Delta$ for which there exists entry larger than $n^{1/2+1/(2p)}$ in magnitude is $O(n^{(2-p)/2})$, which since $p > 1$ is a constant bounded away from 1, is sublinear. Let us call this set with entries larger than $n^{1/2+1/(2p)}$ in magnitude the set $H$ of "heavy" columns; we will not make any guarantees about $H$, rather, we will stuff it into the small set $T$ of columns above on which we will run our earlier $O(k \log k)$-approximation.

For the remaining, non-heavy columns, which constitute almost all of our columns, we show in Lemma 2.5 that $\|\Delta_i\|_1 \le (1 + \epsilon)n$ w.h.p. The reason this is important is that recall to cover some column $i$ by a sample set $I$ of columns, we need $\min_y \|A_I y - A_i\|_1 \le (1 + O(\epsilon))n$. It turns out, as we now explain, that we will get $\min_y \|A_I y - A_i\|_1 \le \|\Delta_i\|_1 + e_i$, where $e_i$ is a quantity which we can control and make $O(\epsilon n)$ by increasing our sample size $I$. Consequently, since $\|\Delta_i\|_1 \le (1+\epsilon)n$, overall we will have $\min_y \|A_I y - A_i\|_1 \le (1 + O(\epsilon))n$, which means that $i$ will be covered. We now explain what $e_i$ is, and why $\min_y \|A_I y - A_i\|_1 \le \|\Delta_i\|_1 + e_i$.

Towards this end, we first explain a key insight in this model. Since the $p$-th moment exists for some real number $p > 1$ (e.g., $p = 1.000001$ suffices), *averaging* helps reduce the noise of fitting a column $A_i$ by subsets of other columns. Namely, we show in Lemma 2.2 that for any $t$ non-heavy column $\Delta_{i_1}, \ldots, \Delta_{i_t}$ of $\Delta$, and any coefficients $\alpha_1, \alpha_2, \ldots, \alpha_t \in [-1, 1]$, $\|\sum_{j=1}^{t} \alpha_j \Delta_{i_j}\|_1 = O(t^{1/p}n)$, that is, since the individual coordinates of the $\Delta_{i_j}$ are zero-mean random variables, their sum *concentrates* as we add up more columns. We do not need bounded variance for this property.

How can we use this averaging property for subset selection? The idea is, instead of sampling a single subset $I$ of $O(k)$ columns and trying to cover each remaining column with this subset as shown in [25], we will sample multiple independent subsets $I_1, I_2, \ldots, I_t$. Each set has size $\text{poly}(k/\epsilon)$ and we will sample at most $\text{poly}(k/\epsilon)$ subsets. By a similar argument of [25], for any given column index $i \in [n]$, for most of these subset $I_j$, we have that $A_i^*/\|\Delta_i\|_1$ can be expressed as a linear combination of columns $A_\ell^*/\|\Delta_\ell\|_1, \ell \in I_j$, via coefficients of absolute value at most 1. Note that this is only true for most $i$ and most $j$; we develop terminology for this in Definitions 2.6,

2.7, 2.8, and 2.9, referring to what we call a *good core*. We quantify what we mean by most $i$ and most $j$ having this property in Lemma 2.11 and Lemma 2.12.

The key though, that drives the analysis, is Lemma 2.10, which shows that $\min_y \|A_i y - A_i\|_1 \leq \|\Delta_i\|_1 + e_i$, where $e_i = O(q^{1/p}/t^{1-1/p} n)$, where $q$ is the size of each $I_j$, and $t$ is the number of different $I_j$. We need $q$ to be at least $k$, just as before, so that we can be guaranteed that when we adjoin a column index $i$ to $I_j$, there is some positive probability that $A_i^*/\|\Delta_i\|_1$ can be expressed as a linear combination of columns $A_\ell^*/\|\Delta_\ell\|_1, \ell \in I_j$, with coefficients of absolute value at most 1. What is different in our noise model though is the division by $t^{1-1/p}$. Since $p > 1$, if we set $t$ to be a large enough $\text{poly}(k/\epsilon)$, then $e_i = O(\epsilon n)$, and then we will have covered $A_i$, as desired. This captures the main property that averaging the linear combinations for expression $A_i^*/\|\Delta_i\|_1$ using different subsets $I_j$ gives us better and better approximations to $A_i^*/\|\Delta_i\|_1$. Of course we need to ensure several properties such as not sampling a heavy column (the averaging in Lemma 2.2 does not apply when this happens), we need to ensure most of the $I_j$ have small-coefficient linear combinations expressing $A_i^*/\|\Delta_i\|_1$, etc. This is handled in our main theorem, Theorem 2.13.

## 2 $\ell_1$-Norm Column Subset Selection

We first present two subroutines.

**Linear regression with $\ell_1$ loss.** The first subroutine needed is an approximate $\ell_1$ linear regression solver. In particular, given a matrix $M \in \mathbb{R}^{n \times d}$, $n$ vectors $b_1, b_2, \cdots, b_n \in \mathbb{R}^n$, and an error parameter $\epsilon \in (0, 1)$, we want to compute $x_1, x_2, \cdots, x_n \in \mathbb{R}^d$ for which $\forall i \in [n]$, we have

$$\|M x_i - b_i\|_1 \leq (1 + \epsilon) \cdot \min_{x \in \mathbb{R}^d} \|M x - b_i\|_1.$$

Furthermore, we also need an estimate $v_i$ of the regression cost $\|M x_i - b_i\|_1$ for each $i \in [n]$ such that $\|M x_i - b_i\|_1 \leq v_i \leq (1 + \epsilon)\|M x_i - b_i\|_1$. Such an $\ell_1$-regression problem can be solved efficiently (see [28] for a survey). The total running time to solve these $n$ regression problems simultaneously is at most $\widetilde{O}(n^2) + n \cdot \text{poly}(d \log n)$, and the success probability is at least 0.999.

**$\ell_1$ Column subset selection for general matrices.** The second subroutine needed is an $\ell_1$-low rank approximation solver for general input matrices, though we allow a large approximation ratio. We use the algorithm proposed by [25] for this purpose. In particular, given an $n \times d$ $(d \leq n)$ matrix $M$ and a rank parameter $k$, the algorithm can output a small set $S \subset [n]$ with size at most $O(k \log n)$, such that

$$\min_{X \in \mathbb{R}^{|S| \times d}} \|M_S X - M\|_1 \leq O(k \log k) \cdot \min_{\text{rank}-k\ B} \|M - B\|_1.$$

Furthermore, the running time is at most $\widetilde{O}(n^2) + n \cdot \text{poly}(k \log n)$, and the success probability is at least 0.999. Now we can present our algorithm, Algorithm 1.

---

**Algorithm 1** $\ell_1$-Low Rank Approximation with Input Assumption

1: **procedure** L1NOISYLOWRANKAPPROX($A \in \mathbb{R}^{n \times n}, k, \epsilon$)  ▷ Theorem 2.13
2:      Sample a set $I$ from $\binom{[n]}{s}$ uniformly at random, where $s = \text{poly}(k/\epsilon)$.
3:      Solve the approximate $\ell_1$-regression problem $\min_{x \in \mathbb{R}^{|I|}} \|A_I x - A_i\|_1$ for each $i \in [n]$, and let $v_i$ be the estimated regression cost.
4:      Compute the set $T = \{i \in [n] \mid v_i \text{ is one of the top } l \text{ largest values among } v_1, v_2, \cdots, v_n\}$, where $l = n/\text{poly}(k/\epsilon)$.
5:      Solve $\ell_1$-column subset selection for $A_T$. Let the solution be $A_Q$.
6:      Solve the approximate $\ell_1$-regression problem $\min_{X \in \mathbb{R}^{(|I|+|Q|) \times n}} \|A_{(I \cup Q)} X - A\|_1$, and let $\widehat{X}$ be the solution. Return $A_{(I \cup Q)}$ and $\widehat{X}$. ▷ $A_{(I \cup Q)}\widehat{X}$ is a good low rank approximation to $A$
7: **end procedure**

---

**Running time.** Uniformly sampling a set $I$ can be done in $\text{poly}(k/\epsilon)$ time. According to our $\ell_1$-regression subroutine, solving $\min_x \|A_I x - A_i\|_1$ for all $i \in [n]$ can be finished in $\widetilde{O}(n^2) + n \cdot \text{poly}(k \log(n)/\epsilon)$ time. We only need sorting to compute the set $T$ which takes $O(n \log n)$ time. By

our second subroutine, the $\ell_1$-column subset selection for $A_T$ will take $\widetilde{O}(n^2) + n \cdot \text{poly}(k \log n)$. The last step only needs an $\ell_1$-regression solver, which takes $\widetilde{O}(n^2) + n \cdot \text{poly}(k \log(n)/\epsilon)$ time. Thus, the overall running time is $\widetilde{O}(n^2) + n \cdot \text{poly}(k \log(n)/\epsilon)$.

The remaining parts in this section will focus on analyzing the correctness of the algorithm.

## 2.1 Properties of the Noise Matrix

Recall that the input matrix $A \in \mathbb{R}^{n \times n}$ can be decomposed as $A^* + \Delta$, where $A^*$ is the ground truth, and $\Delta$ is a random noise matrix. In particular, $A^*$ is an arbitrary rank-$k$ matrix, and $\Delta$ is a random matrix where each entry is an i.i.d. sample drawn from an unknown symmetric distribution. The only assumption on $\Delta$ is that each entry $\Delta_{i,j}$ satisfies $\mathbf{E}[|\Delta_{i,j}|^p] = O(\mathbf{E}[|\Delta_{i,j}|^p])$ for some constant $p > 1$, i.e., the $p$-th moment of the noise distribution is bounded. Without loss of generality, we will suppose $\mathbf{E}[|\Delta_{i,j}|] = 1$, $\mathbf{E}[|\Delta_{i,j}|^p] = O(1)$, and $p \in (1, 2)$ throughout the paper. In this section, we will present some key properties of the noise matrix.

The following lemma provides a lower bound on $\|\Delta\|_1$. Once we have the such lower bound, we can focus on finding a solution for which the approximation cost is at most that lower bound.

**Lemma 2.1** (Lower bound on the noise matrix). *Let $\Delta \in \mathbb{R}^{n \times n}$ be a random matrix where $\Delta_{i,j}$ are i.i.d. samples drawn from a symmetric distribution. Suppose $\mathbf{E}[|\Delta_{i,j}|] = 1$ and $\mathbf{E}[|\Delta_{i,j}|^p] = O(1)$ for some constant $p \in (1, 2)$. Then, $\forall \epsilon \in (0, 1)$ which satisfies $1/\epsilon = n^{o(1)}$, we have*

$$\Pr\left[\|\Delta\|_1 \geq (1 - \epsilon)n^2\right] \geq 1 - e^{-\Theta(n)}.$$

The next lemma shows the main reason why we are able to get a small fitting cost when running regression. Consider a toy example. Suppose we have a target number $a \in \mathbb{R}$, and another $t$ numbers $a + g_1, a + g_2, \cdots, a + g_t \in \mathbb{R}$, where $g_i$ are i.i.d. samples drawn from the standard Gaussian distribution $N(0, 1)$. If we use $a + g_i$ to fit $a$, then the expected cost is $\mathbf{E}[|a + g_i - a|] = \mathbf{E}[|g_i|] = \sqrt{2/\pi}$. However, if we use the average of $a + g_1, a + g_2, \cdots, a + g_t$ to fit $a$, then the expected cost is $\mathbf{E}[|\sum_{i=1}^t g_i|/t]$. Since the $g_i$ are independent, $\sum_{i=1}^t g_i$ is a random Gaussian variable with variance $t$, which means that the above expected cost is $\sqrt{2/\pi}/\sqrt{t}$. Thus the fitting cost is reduced by a factor $\sqrt{t}$. By generalizing the above argument, we obtain the following lemma.

**Lemma 2.2** (Averaging reduces the noise). *Let $\Delta_1, \Delta_2, \cdots, \Delta_t \in \mathbb{R}^n$ be $t$ random vectors. The $\Delta_{i,j}$ are i.i.d. symmetric random variables with $\mathbf{E}[|\Delta_{i,j}|] = 1$ and $\mathbf{E}[|\Delta_{i,j}|^p] = O(1)$ for some constant $p \in (1, 2)$. Let $\alpha_1, \alpha_2, \cdots, \alpha_t \in [-1, 1]$ be $t$ real numbers. Conditioned on $\forall i \in [n], j \in [t], |\Delta_{i,j}| \leq n^{1/2 + 1/(2p)}$, with probability at least $1 - 2^{-n^{\Theta(1)}}$,*

$$\left\|\sum_{i=1}^t \alpha_i \Delta_i\right\|_1 \leq O(t^{1/p}n).$$

The above lemma needs a condition that each entry in the noise column should not be too large. Fortunately, we can show that most of the (noise) columns do not have any large entry.

**Lemma 2.3** (Only a small number of columns have large entries). *Let $\Delta \in \mathbb{R}^{n \times n}$ be a random matrix where the $\Delta_{i,j}$ are i.i.d. symmetric random variables with $\mathbf{E}[|\Delta_{i,j}|] = 1$ and $\mathbf{E}[|\Delta_{i,j}|^p] = O(1)$ for some constant $p \in (1, 2)$. Let*

$$H = \{j \in [n] \mid \exists i \in [n], |\Delta_{i,j}| > n^{1/2 + 1/(2p)}\}.$$

*Then with probability at least $0.999$ $|H| \leq O(n^{1 - (p-1)/2})$.*

The following lemma shows that any small subset of the columns of the noise matrix $\Delta$ cannot contribute too much to the overall error. By combining with the previous lemma, the entrywise $\ell_1$ cost of all columns containing large entries can be bounded.

**Lemma 2.4.** *Let $\Delta \in \mathbb{R}^{n \times n}$ be a random matrix where $\Delta_{i,j}$ are i.i.d. symmetric random variables with $\mathbf{E}[|\Delta_{i,j}|] = 1$ and $\mathbf{E}[|\Delta_{i,j}|^p] = O(1)$ for some constant $p \in (1, 2)$. Let $\epsilon \in (0, 1)$ satisfy $1/\epsilon = n^{o(1)}$. Let $r \geq (1/\epsilon)^{1 + 1/(p-1)}$. Then, with probability at least $.999$, $\forall S \subset [n]$ with $|S| \leq n/r$, $\sum_{j \in S} \|\Delta_j\|_1 = O(\epsilon n^2)$.*

We say a (noise) column is good if it does not have a large entry. We can show that, with high probability, the entry-wise $\ell_1$ cost of a good (noise) column is small.

**Lemma 2.5** (Cost of good noise columns). *Let $\Delta \in \mathbb{R}^n$ be a random vector where $\Delta_i$ are i.i.d. symmetric random variables with $\mathbf{E}[|\Delta_i|] = 1$ and $\mathbf{E}[|\Delta_i|^p] = O(1)$ for some constant $p \in (1, 2)$. Let $\epsilon \in (0, 1)$ satisfy $1/\epsilon = n^{o(1)}$. If $\forall i \in [n], |\Delta_i| \le n^{1/2+1/(2p)}$, then with probability at least $1 - 2^{-n^{\Theta(1)}}, \|\Delta\|_1 \le (1 + \epsilon)n$.*

## 2.2 Definition of Tuples and Cores

In this section, we provide some basic definitions, e.g., of a tuple, a good tuple, the core of a tuple, and a coefficients tuple. These definitions will be heavily used later when we analyze the correctness of our algorithm.

Before we present the definitions, we introduce a notion $R_{A^*}(S)$. Given a matrix $A^* \in \mathbb{R}^{n_1 \times n_2}$, for a set $S \subseteq [n_2]$, we define

$$R_{A^*}(S) := \arg \max_{P: P \subseteq S} \left\{ \left| \det\left( (A^*)^Q_P \right) \right| \, \middle| \, |P| = |Q| = \text{rank}(A^*_S), Q \subseteq [n_1] \right\},$$

where for a squared matrix $C$, $\det(C)$ denotes the determinant of $C$. The above maximum is over both $P$ and $Q$ while $R_{A^*}(S)$ only takes the value of the corresponding $P$. By Cramer's rule, if we use the columns of $A^*$ with index in the set $R_{A^*}(S)$ to fit any column of $A^*$ with index in the set $S$, the absolute value of any fitting coefficient will be at most 1. The use of Cramer's rule is as follows. Consider a rank $k$ matrix $M \in \mathbb{R}^{n \times (k+1)}$. Let $P \subseteq [k+1], Q \subseteq [n], |P| = |Q| = k$ be such that $|\det(M^Q_P)|$ is maximized. Since $M$ has rank $k$, we know $\det(M^Q_P) \ne 0$ and thus the columns of $M_P$ are independent. Let $i \in [k+1] \setminus P$. Then the linear equation $M_P x = M_i$ is feasible and there is a unique solution $x$. Furthermore, by Cramer's rule $x_j = \det(M^Q_{[k+1] \setminus \{j\}})/\det(M^Q_P)$. Since $|\det(M^Q_P)| \ge |\det(M^Q_{[k+1] \setminus \{j\}})|$, we have $\|x\|_\infty \le 1$.

Small fitting coefficients are good since they will not increase the noise by too much. For example, suppose $A^*_i = A^*_S x$ and $\|x\|_\infty \le 1$, i.e., the $i$-th column can be fit by the columns with indices in the set $S$ and the fitting coefficients $x \in \mathbb{R}^{|S|}$ are small. If we use the noisy columns of $A^*_S + \Delta_S$ to fit the noisy column $A^*_i + \Delta_i$, then the fitting cost is at most $\|(A^*_S + \Delta_S)x - (A^*_i + \Delta_i)\|_1 \le \|\Delta_i\|_1 + \|\Delta_S x\|_1$. Since $\|x\|_\infty \le 1$, it is possible to give a good upper bound for $\|\Delta_S x\|_1$.

**Definition 2.6** (Tuple). *A $(q, t, n)-$tuple is defined to be $(S_1, S_2, \cdots, S_t, i)$, where $\forall j \in [t], S_j \subset [n]$ with $|S_j| = q$. Let $S = \bigcup_{j=1}^t S_j$. Then $|S| = qt$, i.e., $S_1, S_2, \cdots, S_t$ are disjoint. Furthermore, $i \in [n]$ and $i \notin S$. For simplicity, we use $(S_{[t]}, i)$ to denote $(S_1, S_2, \cdots, S_t, i)$.*

We next provide the definition of a good tuple.

**Definition 2.7** (Good tuple). *Given a rank-$k$ matrix $A^* \in \mathbb{R}^{n \times n}$, an $(A^*, q, t, \alpha)$-good tuple is a $(q, t, n)$-tuple $(S_{[t]}, i)$ which satisfies*

$$|\{j \in [t] \mid i \notin R_{A^*}(S_j \cup \{i\})\}| \ge \alpha \cdot t.$$

We need the definition of the core of a tuple.

**Definition 2.8** (Core of a tuple). *The core of $(S_{[t]}, i)$ is defined to be the set*

$$\{j \in [t] \mid i \notin R_{A^*}(S_j \cup \{i\})\}.$$

We define a coefficients tuple as follows.

**Definition 2.9** (Coefficients tuple). *Given a rank-$k$ matrix $A^* \in \mathbb{R}^{n \times n}$, let $(S_{[t]}, i)$ be an $(A^*, q, t, \alpha)$-good tuple. Let $C$ be the core of $(S_{[t]}, i)$. A coefficients tuple corresponding to $(S_{[t]}, i)$ is defined to be $(x_1, x_2, \cdots, x_t)$ where $\forall j \in [t], x_j \in \mathbb{R}^q$. The vector $x_j \in \mathbb{R}^q$ satisfies: $x_j = 0$ if $j \in [t] \backslash C$, while $A^*_{S_j} x_j = A^*_i$ and $\|x_j\|_\infty \le 1$, if $j \in C$. To guarantee the coefficients tuple is unique, we restrict each vector $x_j \in \mathbb{R}^q$ to be one that has the minimum lexicographic order.*

## 2.3 Properties of a Good Tuple and a Coefficients Tuple

Consider a good tuple $(S_1, S_2, \cdots, S_t, i)$. By the definition of a good tuple, the size of the core $C$ of the tuple is large. For each $j \in C$, the coefficients $x_j$ of using $A^*_{S_j}$ to fit $A^*_i$ should have absolute value at most 1. Now consider the noisy setting. As discussed in the previous section, using $A_{S_j}$ to fit $A_i$ has cost at most $\|\Delta_i\|_1 + \|\Delta_{S_j} x_j\|_1$. Although $\|\Delta_{S_j} x_j\|_1$ has a good upper bound, it is not small enough. To further reduce the $\ell_1$ fitting cost, we can now apply the averaging argument (Lemma 2.2) over all the fitting choices corresponding to $C$. Formally, we have the following lemma.

**Lemma 2.10** (Good tuples imply low fitting cost). *Suppose we are given a matrix $A \in \mathbb{R}^{n \times n}$ which satisfies $A = A^* + \Delta$, where $A^* \in \mathbb{R}^{n \times n}$ has rank $k$. Here $\Delta \in \mathbb{R}^{n \times n}$ is a random matrix where $\Delta_{i,j}$ are i.i.d. symmetric random variables with $\mathbf{E}[|\Delta_{i,j}|] = 1$ and $\mathbf{E}[|\Delta_{i,j}|^p] = O(1)$ for some constant $p \in (1, 2)$. Let $H \subset [n]$ be defined as follows:*

$$H = \left\{ j \in [n] \; \middle| \; \exists i \in [n], |\Delta_{i,j}| > n^{1/2 + 1/(2p)} \right\}.$$

*Let $q, t \leq n^{o(1)}$. Then, with probability at least $1 - 2^{-n^{\Theta(1)}}$, for all $(A^*, q, t, 1/2)$-good tuples $(S_1, S_2, \cdots, S_t, i)$ which satisfy $H \cap \left( \bigcup_{j=1}^t S_j \right) = \emptyset$, we have*

$$\min_{y \in \mathbb{R}^{qt}} \left\| A_{\{\bigcup_{j=1}^t S_j\}} y - A_i \right\|_1 \leq \left\| \frac{1}{|C|} \sum_{j=1}^t A_{S_j} x_j - A_i \right\|_1 \leq \|\Delta_i\|_1 + O(q^{1/p}/t^{1-1/p}n),$$

*where $C$ is the core of $(S_1, S_2, \cdots, S_t, i)$, and $(x_1, x_2, \cdots, x_t)$ is the coefficients tuple corresponding to $(S_1, S_2, \cdots, S_t, i)$.*

We next show that if we choose columns randomly, it is easy to find a good tuple.

**Lemma 2.11.** *Given a rank-$k$ matrix $A^* \in \mathbb{R}^{n \times n}$, let $q > 10k, t > 0$. Let $I = \{i_1, i_2, \cdots, i_{qt+1}\}$ be a subset drawn uniformly at random from $\binom{[n]}{qt+1}$. Let $\pi : I \to I$ be a random permutation of $qt + 1$ elements. $\forall j \in [t]$, let*

$$S_j = \left\{ i_{\pi((j-1)q+1)}, i_{\pi((j-1)q+2)}, \cdots, i_{\pi((j-1)q+q)} \right\}.$$

*We use $i$ to denote $i_{\pi(qt+1)}$. With probability $\geq 1 - 2k/q$, $(S_1, S_2, \cdots, S_t, i)$ is an $(A^*, q, t, 1/2)$−good tuple.*

Lemma 2.11 implies that if we randomly choose $S_1, S_2, \cdots, S_t$, then with high probability, there are many choices of $i \in [n]$, such that $(S_1, S_2, \cdots, S_t, i)$ is a good tuple. Precisely, we can show the following.

**Lemma 2.12.** *Given a rank-$k$ matrix $A^* \in \mathbb{R}^{n \times n}$, let $q > 10k, t > 0$. Let $I = \{i_1, i_2, \cdots, i_{qt}\}$ be a random subset uniformly drawn from $\binom{[n]}{qt}$. Let $\pi$ be a random permutation of $qt$ elements. $\forall j \in [t]$, we define $S_j$ as follows:*

$$S_j = \left\{ i_{\pi((j-1)q+1)}, i_{\pi((j-1)q+2)}, \cdots, i_{\pi((j-1)q+q)} \right\}.$$

*Then with probability at least $2k/q$,*

$$\left| \left\{ i \in [n] \setminus I \; \middle| \; (S_1, S_2, \cdots, S_t, i) \text{ is an } (A^*, q, t, 1/2)-\text{good tuple} \right\} \right| \geq (1 - 4k/q)(n - qt).$$

## 2.4 Main Result

Now we are able to put all ingredients together to prove our main theorem, Theorem 2.13.

**Theorem 2.13** (Formal version of Theorem 1.1). *Suppose we are given a matrix $A = A^* + \Delta \in \mathbb{R}^{n \times n}$, where $\text{rank}(A^*) = k$ for $k = n^{o(1)}$, and $\Delta$ is a random matrix for which the $\Delta_{i,j}$ are i.i.d. symmetric random variables with $\mathbf{E}[|\Delta_{i,j}|] = 1$ and $\mathbf{E}[|\Delta_{i,j}|^p] = O(1)$ for some constant $p \in (1, 2)$. Let $\epsilon \in (0, 1/2)$ satisfy $1/\epsilon = n^{o(1)}$. There is an $\widetilde{O}(n^2 + n \,\text{poly}(k/\epsilon))$ time algorithm (Algorithm 1) which can output a subset $S \in [n]$ with $|S| \leq \text{poly}(k/\epsilon) + O(k \log n)$ for which*

$$\min_{X \in \mathbb{R}^{|S| \times n}} \|A_S X - A\|_1 \leq (1 + \epsilon)\|\Delta\|_1,$$

*holds with probability at least $99/100$.*

*Proof.* We discussed the running time at the beginning of Section 2. Next, we turn to correctness. Let $q = \Omega\left(\frac{k(k\log k)^{1+\frac{1}{p-1}}}{\epsilon^{1+\frac{1}{p-1}}}\right), t = \frac{q^{\frac{1}{p-1}}}{\epsilon^{1+\frac{1}{p-1}}}$. Let $r = \Theta(q/k)$. Let

$$I_1 = \left\{i_1^{(1)}, i_2^{(1)}, \cdots, i_{qt}^{(1)}\right\}, I_2 = \left\{i_1^{(2)}, i_2^{(2)}, \cdots, i_{qt}^{(2)}\right\}, \cdots, I_r = \left\{i_1^{(r)}, i_2^{(r)}, \cdots, i_{qt}^{(r)}\right\},$$

be $r$ independent subsets drawn uniformly at random from $\binom{[n]}{qt}$. Let $I = \bigcup_{s\in[r]} I_s$, which is the same as that in Algorithm 1. Let $\pi_1, \pi_2, \cdots, \pi_r$ be $r$ independent random permutations of $qt$ elements. Due to Lemma 2.12 and a Chernoff bound, with probability at least .999, $\exists s \in [r]$,

$$\left|\left\{i \in [n] \setminus I_s \mid (S_1, S_2, \cdots, S_t, i) \text{ is an } (A^*, q, t, 1/2)-\text{good tuple }\right\}\right| \geq (1 - 4k/q)(n - qt)$$

where

$$S_j = \left\{i_{\pi_s((j-1)q+1)}^{(s)}, i_{\pi_s((j-1)q+2)}^{(s)}, \cdots, i_{\pi_s((j-1)q+q)}^{(s)}\right\}, \forall j \in [t].$$

Let set $H \subset [n]$ be defined as follows:

$$H = \{j \in [n] \mid \exists i \in [n], |\Delta_{i,j}| > n^{1/2+1/(2p)}\}.$$

Then due to Lemma 2.3, with probability at least $0.999$, $|H| \leq O(n^{1-(p-1)/2})$. Thus, for $j \in [r]$, the probability that $H \cap I_j \neq \emptyset$ is at most $O(qt \cdot n^{1-(p-1)/2}/(n - qt)) = 1/n^{\Omega(1)}$. By taking a union bound over all $j \in [r]$, with probability at least $1 - 1/n^{\Omega(1)}$, $\forall j \in [r], I_j \cap H = \emptyset$. Thus, we can condition on $I_s \cap H = \emptyset$. Due to Lemma 2.10 and $q^{1/p}/t^{1-1/p} = \epsilon$,

$$\left|\left\{i \in [n] \setminus I_s \mid \min_{y\in\mathbb{R}^{qt}} \|A_{I_s}y - A_i\|_1 \leq \|\Delta_i\|_1 + O(\epsilon n)\right\}\right| \geq (1 - 4k/q)(n - qt).$$

Due to Lemma 2.5 and a union bound over all $i \in [n] \setminus H$, with probability at least .999, $\forall i \notin H, \|\Delta_i\| \leq (1 + \epsilon)n$. Thus,

$$\left|\left\{i \in [n] \setminus I_s \mid \min_{y\in\mathbb{R}^{qt}} \|A_{I_s}y - A_i\|_1 \leq (1 + O(\epsilon))n\right\}\right| \geq (1 - 4k/q)(n - qt) - |H|.$$

Let

$$T' = [n] \setminus \left\{i \in [n] \mid \min_{y\in\mathbb{R}^{qt}} \|A_{I_s}y - A_i\|_1 \leq (1 + O(\epsilon))n\right\}.$$

Then $|T'| \leq O(kn/q + n^{1-(p-1)/2}) = O(kn/q) = O((\epsilon/(k\log k))^{1+1/(p-1)}n)$. By our selection of $T$ in algorithm 1, $T'$ should be a subset of $T$. Due to Lemma 2.4, with probability at least .999, $\|\Delta_T\|_1 \leq O(\epsilon n^2/(k\log k))$. By our second subroutine mentioned at the beginning of Section 2 it can find a set $Q \subset [n]$ with $|Q| = O(k\log n)$ such that $\min_{X\in\mathbb{R}^{|Q|\times|T|}} \|A_Q X - A_T\|_1 \leq O(k\log k)\|\Delta_T\|_1 \leq O(\epsilon n^2)$. Thus, we have $\min_{X\in\mathbb{R}^{(|Q|+q\cdot t\cdot r)\times n}} \|A_{(Q\cup I)}X - A\|_1 \leq \min_{X_1\in\mathbb{R}^{(q\cdot t\cdot r)\times n}} \|A_I X_1 - A_{[n]\setminus T}\|_1 + \min_{X_2\in\mathbb{R}^{|Q|\times n}} \|A_Q X_2 - A_T\|_1 \leq (1 + O(\epsilon))n^2$. Due to Lemma 2.1, with probability at least .999, $\|\Delta\|_1 \geq (1 - \epsilon)n^2$, and thus $\min_{X\in\mathbb{R}^{(|Q|+q\cdot t\cdot r)\times n}} \|A_{(Q\cup I)}X - A\|_1 \leq (1 + O(\epsilon))\|\Delta\|_1$. □

## 3 Experiments

The take-home message from our theoretical analysis is that although the noise distribution may be heavy-tailed, if the $p$-th $(p > 1)$ moment of the distribution exists, averaging the noise may reduce the noise. In the spirit of averaging, we found that taking a median works a bit better in practice. Inspired by our theoretical analysis, we propose a simple heuristic algorithm (Algorithm 2) which can output a rank-$k$ solution. We tested Algorithm 2 on both synthetic and real datasets.

**Datasets.** For each rank-$k$ experiment, we chose a high rank matrix $\widehat{A} \in \mathbb{R}^{n\times d}$, applied top-$k$ SVD to $\widehat{A}$ and obtained a rank-$k$ matrix $A^*$ as our ground truth matrix. For our synthetic data experiments, the matrix $\widehat{A} \in \mathbb{R}^{500\times500}$ was generated at random, where each entry was drawn uniformly from

---
**Algorithm 2** Median Heuristic
---
1: **procedure** L1NOISYLOWRANKAPPROXHEU($A \in \mathbb{R}^{n \times d}, k \geq 1$)
2:      Sample a set $I = \{i_1, i_2, \cdots, i_{sk}\}$ from $\binom{[n]}{sk}$ uniformly at random.
3:      Compute $B \in \mathbb{R}^{n \times k}$ s.t., for $t \in [n], q \in [k]$, $B_{t,q} = \text{median}(A_{t,i_{s(q-1)+1}}, \cdots, A_{t,i_{sq}})$.
4:      Solve $\min_{X \in \mathbb{R}^{k \times d}} \|BX - A\|_1$ and let the solution be $X^*$. Output $BX^*$.
5: **end procedure**
---

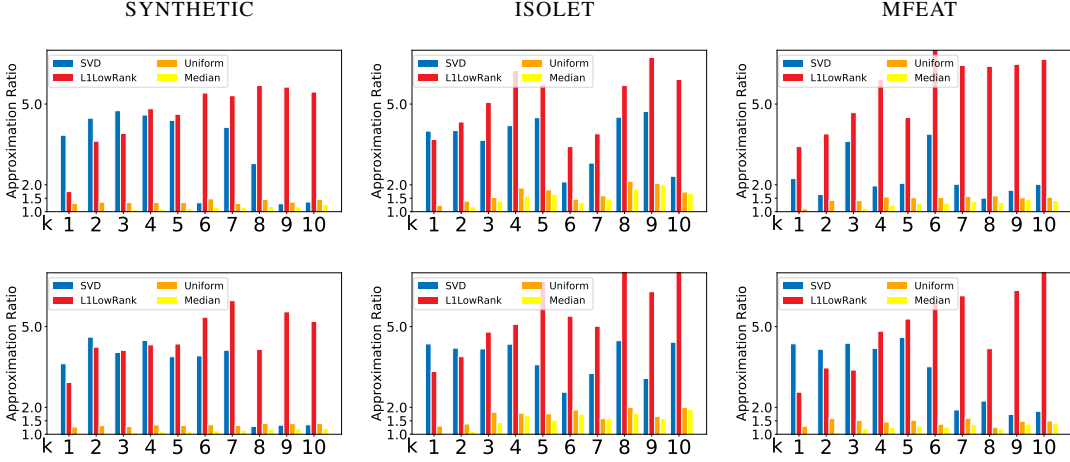

Figure 1: **Empirical results.** The noise distributions of the experiments in the first row are from a 1.1-stable distribution. The noise distributions corresponding to the second row are the 1.1-th root of a Cauchy distribution. The blue, red, orange and yellow bar denote SVD, the entrywise $\ell_1$-norm low rank algorithm in [24], the uniform $k$-column subset sampling algorithm in [25], and Algorithm 2 respectively.

$\{0, 1, \cdots, 9\}$. For real datasets, we chose *isolet*[3] ($617 \times 1559$) or *mfeat*[4] ($651 \times 2000$) as $\widehat{A}$ [29]. We tested two different noise distributions. One distribution is the standard Lévy 1.1-stable distribution [30]. Another distribution is constructed from the standard Cauchy distribution, i.e., to draw a sample from the constructed distribution, we draw a sample from the Cauchy distribution, keep the sign unchanged, and take the $\frac{1}{1.1}$-th power of the absolute value. Notice that both distributions have bounded 1.1-th moment, but do not have a $p$-th moment for any $p > 1.1$. To construct the noise matrix $\Delta \in \mathbb{R}^{n \times d}$, we drew a matrix $\widehat{\Delta}$ where each entry is an i.i.d. sample from one of the two noise distributions, and then scaled the noise: $\Delta = \widehat{\Delta} \cdot \frac{\|A^*\|_1}{20 \cdot n \cdot d}$. We set $A = A^* + \Delta$ as the input.

**Methodologies.** We compare Algorithm 2 with SVD, $\text{poly}(k, \log n)$-approximate entrywise $\ell_1$ low rank approximation [24], and uniform $k$-column subset sampling [25][5]. For Algorithm 2, we set $s = \min(50, \lfloor n/k \rfloor)$. For all of algorithms we repeated the experiment the same number of times and compared the best solution obtained by each algorithm. We report the approximation ratio $\|B - A\|_1 / \|\Delta\|_1$ for each algorithm, where $B \in \mathbb{R}^{n \times d}$ is the output rank-$k$ matrix. The results are shown in Figure 1. As shown in the figure, Algorithm 2 outperformed all of the other algorithms.

**Acknowledgments.** David P. Woodruff was supported in part by Office of Naval Research (ONR) grant N00014- 18-1-2562. Part of this work was done while he was visiting the Simons Institute for the Theory of Computing. Peilin Zhong is supported in part by NSF grants (CCF-1703925, CCF-1421161, CCF-1714818, CCF-1617955 and CCF-1740833), Simons Foundation (#491119 to Alexandr Andoni), Google Research Award and a Google Ph.D. fellowship. Part of this work was done while Zhao Song and Peilin Zhong were interns at IBM Research - Almaden and while Zhao Song was visiting the Simons Institute for the Theory of Computing.

## Footnotes

[2]We use the notation $\widetilde{O}(f) := O(f \cdot \log^{O(1)} f)$.

[3]https://archive.ics.uci.edu/ml/datasets/isolet

[4]https://archive.ics.uci.edu/ml/datasets/Multiple+Features

[5]We chose to compare with [24, 25] due to their theoretical guarantees. Though the uniform $k$-column subset sampling described in the experiments of [25] is a heuristic algorithm, it is inspired by their theoretical algorithm.

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
