[Supplementary Material]

# A    Missing Proofs in Section 2

## A.1    Proof of Lemma 2.1

*Proof.* Let $Z \in \mathbb{R}^{n \times n}$ be a random matrix. For each $i, j \in [n]$, define random variable $Z_{i,j}$ as

$$Z_{i,j} = \begin{cases} |\Delta_{i,j}|, & \text{if } |\Delta_{i,j}| \leq n; \\ n, & \text{otherwise.} \end{cases}$$

For $i, j \in [n]$, by Markov's inequality, we have

$$\Pr[|\Delta_{i,j}| \geq n] = \Pr[|\Delta_{i,j}|^p \geq n^p] \leq \mathbf{E}[|\Delta_{i,j}|^p]/n^p = O(1/n^p). \tag{2}$$

Notice that

$$\mathbf{E}[|\Delta_{i,j}|^p] = \int_0^n x^p f(x) \mathrm{d}x + \int_n^\infty x^p f(x) \mathrm{d}x = O(1)$$

where $f(x)$ is the probability density function of $|\Delta_{i,j}|$. Thus we have

$$\int_n^\infty x f(x) \mathrm{d}x \leq \int_n^\infty x^p/n^{p-1} \cdot f(x) \mathrm{d}x = O(1/n^{p-1}).$$

Because $\mathbf{E}[|\Delta_{i,j}|] = 1$, we have

$$\int_0^\infty x f(x) \mathrm{d}x = \mathbf{E}[|\Delta_{i,j}|] - \int_n^\infty x f(x) \mathrm{d}x \geq 1 - O(1/n^{p-1}). \tag{3}$$

By Equation (3), we have

$$\mathbf{E}[Z_{i,j}] = \int_0^n x f(x) \mathrm{d}x + n \cdot \Pr[|\Delta_{i,j}| \geq n] \geq \int_0^n x f(x) \mathrm{d}x \geq 1 - O(1/n^{p-1}).$$

By Equation (2) and $E[|\Delta_{i,j}|^p] \leq O(1)$, we have

$$\mathbf{E}[Z_{i,j}^2] = \int_0^n x^2 f(x) \mathrm{d}x + n^2 \Pr[|\Delta_{i,j}| \geq n] \leq O(n^{2-p}) + O(n^{2-p}) = O(n^{2-p}).$$

By the inequality of [31],

$$\Pr[\mathbf{E}[\|Z\|_1] - \|Z\|_1 \geq \epsilon \, \mathbf{E}[\|Z\|_1]/2] \leq \exp\left(\frac{-\epsilon^2 \, \mathbf{E}[\|Z\|_1]^2/4}{2 \sum_{i,j} \mathbf{E}[Z_{i,j}^2]}\right)$$

$$\leq \exp\left(\frac{-\epsilon^2(n^2 - O(n^{3-p}))^2/4}{2n^2 \cdot O(n^{2-p})}\right)$$

$$\leq e^{-\Theta(n)}$$

Thus with probability at least $1 - e^{-\Theta(n)}$, $\|Z\|_1 \geq (1 - \epsilon/2) \, \mathbf{E}[\|Z\|_1] \geq (1 - \epsilon)n^2$ where the last inequality follows by $\mathbf{E}[\|Z\|_1 \geq n^2 - O(n^{3-p})]$ and $1/\epsilon = n^{o(1)}$. Since $\|\Delta\|_1 \geq \|Z\|_1$, we complete the proof. $\qquad \square$

## A.2    Proof of Lemma 2.2

*Proof.* Let $Z \in \mathbb{R}^{n \times t}$ be a random matrix where $Z_{i,j}$ are i.i.d. random variables with probability density function:

$$g(x) = \begin{cases} f(x)/\Pr[|\Delta_{1,1}| \leq n^{1/2+1/(2p)}], & \text{if } |x| \leq n^{1/2+1/(2p)}; \\ 0, & \text{otherwise.} \end{cases}$$

where $f(x)$ is the probability density function of $\Delta_{1,1}$. (Note that in the above equation, $\Pr[|\Delta_{1,1}| \leq n^{1/2+1/(2p)}] > 0$.) Now, we have $\forall a \geq 0$,

$$\Pr\left[\left\|\sum_{j=1}^t \alpha_j \Delta_j\right\|_1 \leq a \,\middle|\, \forall i \in [n], j \in [t], |\Delta_{i,j}| \leq n^{1/2+1/(2p)}\right] = \Pr\left[\left\|\sum_{j=1}^t \alpha_j Z_j\right\|_1 \leq a\right].$$

Now we look at the $i$-th row of $\sum_{j=1}^{t} \alpha_j Z_j$. We have

$$
\mathbf{E}\left[\left|\sum_{j=1}^{t} \alpha_j Z_{i,j}\right|\right] = \left(\mathbf{E}\left[\left|\sum_{j=1}^{t} \alpha_j Z_{i,j}\right|\right]^p\right)^{1/p}
$$

$$
\leq \mathbf{E}\left[\left|\sum_{j=1}^{t} \alpha_j Z_{i,j}\right|^p\right]^{1/p}
$$

$$
\leq \mathbf{E}\left[\left(\left(\sum_{j=1}^{t} \alpha_j^2 Z_{i,j}^2\right)^{1/2}\right)^p\right]^{1/p}
$$

$$
\leq \mathbf{E}\left[\sum_{j=1}^{t} |\alpha_j Z_{i,j}|^p\right]^{1/p}
$$

$$
\leq \left(\sum_{j=1}^{t} \mathbf{E}[|\alpha_j Z_{i,j}|^p]\right)^{1/p}
$$

$$
\leq \left(\sum_{j=1}^{t} \mathbf{E}[|Z_{i,j}|^p]\right)^{1/p}
$$

$$
\leq O(t^{1/p}), \tag{4}
$$

where the first inequality follows by Jensen's inequality, the second inequality follows by Remark 3 of [32], the third inequality follows by $\|x\|_2 \leq \|x\|_p$ for $p < 2$, the fourth inequality follows by $|\alpha_j| \leq 1$, the fifth inequality follows by $\mathbf{E}[|Z_{i,j}|^p] = \mathbf{E}[|\Delta_{i,j}|^p \mid |\Delta_{1,1}| \leq n^{1/2+1/(2p)}] \leq \mathbf{E}[|\Delta_{i,j}|^p] = O(1)$. For the second moment, we have

$$
\mathbf{E}\left[\left|\sum_{j=1}^{t} \alpha_j Z_{i,j}\right|^2\right] = \sum_{j=1}^{t} \mathbf{E}\left[\alpha_j^2 Z_{i,j}^2\right] + \sum_{j \neq k} \mathbf{E}[\alpha_j \alpha_k Z_{i,j} Z_{i,k}]
$$

$$
= \sum_{j=1}^{t} \alpha_j^2 \, \mathbf{E}\left[Z_{i,j}^2\right] + \sum_{j \neq k} \alpha_j \alpha_k \, \mathbf{E}[Z_{i,j}] \, \mathbf{E}[Z_{i,k}]
$$

$$
\leq \sum_{j=1}^{t} \mathbf{E}\left[Z_{i,j}^2\right]
$$

$$
= t \cdot 2 \int_0^{n^{1/2+1/(2p)}} x^2 f(x)/\Pr\left[|\Delta_{i,j}| \leq n^{1/2+1/(2p)}\right] \mathrm{d}x
$$

$$
\leq 2t/\Pr\left[|\Delta_{i,j}| \leq n^{1/2+1/(2p)}\right] \cdot (n^{1/2+1/(2p)})^{2-p} \int_0^{n^{1/2+1/(2p)}} x^p f(x)\mathrm{d}x
$$

$$
\leq O(tn^{2-p}), \tag{5}
$$

where the second inequality follows by independence of $Z_{i,j}$ and $Z_{i,k}$. The first inequality follows by $|\alpha_j| \leq 1$ and $\mathbf{E}[Z_{i,j}] = \mathbf{E}[Z_{i,k}] = 0$. The third equality follows by the probability density function of $Z_{i,j}$. The second inequality follows by $x^{2-p} \leq (n^{1/2+1/(2p)})^{2-p}$ when $0 \leq x \leq n^{1/2+1/(2p)}$. The last inequality follows by $\mathbf{E}[|\Delta_{i,j}|^p] = O(1), p > 1$ and $\Pr[|\Delta_{i,j}| \leq n^{1/2+1/(2p)}] \geq 1 - \mathbf{E}[|\Delta_{i,j}|^p]/(n^{1/2+1/(2p)})^p = 1 - O(1/n^{p/2+1/2}) \geq 1/2$.

For $i \in [n]$, define $X_i = |\sum_{j=1}^{t} \alpha_j Z_{i,j}|$. Then, by Bernstein's inequality

$$
\Pr\left[\left|\left\|\sum_{j=1}^{t} \alpha_j Z_j\right\|_1 - \mathbf{E}\left[\left\|\sum_{j=1}^{t} \alpha_j Z_j\right\|_1\right]\right| \geq 0.5 t^{1/p} n\right]
$$

$$
= \Pr\left[\sum_{i=1}^{n} X_i - \mathbf{E}\left[\sum_{i=1}^{n} X_i\right] \geq 0.5 t^{1/p} n\right]
$$

$$
\leq \exp\left(-\frac{0.5 \cdot 0.5^2 t^{2/p} n^2}{\sum_{i=1}^{n} \mathbf{E}[X_i^2] + \frac{1}{3} n^{1/2 + 1/(2p)} \cdot 0.5 t^{1/p} n}\right)
$$

$$
\leq e^{-n^{\Theta(1)}}.
$$

The last inequality follows by Equation (5). According to Equation (4), with probability at least $1 - e^{-n^{\Theta(1)}}$,

$$
\left\|\sum_{j=1}^{t} \alpha_j Z_j\right\|_1 \leq \mathbf{E}\left[\left\|\sum_{j=1}^{t} \alpha_j Z_j\right\|_1\right] + 0.5 t^{1/p} n \leq O(t^{1/p} n).
$$

$\square$

### A.3 Proof of Lemma 2.3

*Proof.* For $i, j \in [n]$, we have

$$
\Pr\left[|\Delta_{i,j}| > n^{1/2 + 1/(2p)}\right] = \Pr\left[|\Delta_{i,j}|^p > n^{p/2 + 1/2}\right] \leq \mathbf{E}\left[|\Delta_{i,j}|^p\right]/n^{p/2 + 1/2} \leq O(1/n^{p/2 + 1/2}).
$$

For column $j$, by taking a union bound,

$$
\Pr[j \in H] = \Pr\left[\exists i \in [n], |\Delta_{i,j}| > n^{1/2 + 1/(2p)}\right] \leq O(1/n^{p/2 - 1/2}).
$$

Thus, $\mathbf{E}[|H|] \leq O(n^{1 - (p-1)/2})$. By applying Markov's inequality, we complete the proof. $\square$

### A.4 Proof of Lemma 2.4

*Proof.* For $l \in \mathbb{N}_{\geq 0}$, define $G_l = \{j \mid \|\Delta_j\|_1 \in (n \cdot 2^l, n \cdot 2^{l+1}]\}$. We have

$$
\mathbf{E}[|G_l|] \leq \sum_{j=1}^{n} \Pr\left[\|\Delta_j\|_1 \geq n \cdot 2^l\right]
$$

$$
= n \Pr\left[\|\Delta_1\|_1 \geq n \cdot 2^l\right]
$$

$$
\leq n \Pr\left[n^{1-1/p}\|\Delta_1\|_p \geq n \cdot 2^l\right]
$$

$$
= n \Pr\left[n^{p-1}\|\Delta_1\|_p^p \geq n^p \cdot 2^{lp}\right]
$$

$$
\leq n \mathbf{E}\left[n^{p-1}\|\Delta_1\|_p^p\right]/(n^p \cdot 2^{lp})
$$

$$
\leq O(n/2^{lp}).
$$

The first inequality follows by the definition of $G_l$. The second inequality follows since $\forall x \in \mathbb{R}^n, \|x\|_1 \leq n^{1-1/p}\|x\|_p$. The third inequality follows by Markov's inequality. The last inequality follows since $\forall i, j \in [n], \mathbf{E}[|\Delta_{i,j}|^p] = O(1)$.

Let $l^* \in \mathbb{N}_{\geq 0}$ satisfy $2^{l^*} < \epsilon r$ and $2^{l^*+1} \geq \epsilon r$. We have

$$
\begin{aligned}
\mathbf{E}\left[\sum_{j:\|\Delta_j\|_1 \geq n2^{l^*}} \|\Delta_j\|_1\right] &\leq \mathbf{E}\left[\sum_{l=l^*}^{\infty} |G_l| \cdot n2^{l+1}\right] = \sum_{l=l^*}^{\infty} \mathbf{E}[|G_l|] \cdot n2^{l+1} \\
&\leq \sum_{l=l^*}^{\infty} O(n/2^{lp}) \cdot n2^{l+1} = \sum_{l=l^*}^{\infty} O(n^2/2^{l(p-1)}) \\
&= O(n^2/2^{l^*(p-1)}) = O(n^2/(\epsilon r)^{p-1}) \\
&= O(\epsilon n^2).
\end{aligned}
$$

By Markov's inequality, with probability at least $.999$, $\sum_{j:\|\Delta_j\|_1 \geq n2^{l^*}} \|\Delta_j\|_1 \leq O(\epsilon n^2)$. Conditioned on $\sum_{j:\|\Delta_j\|_1 \geq n2^{l^*}} \|\Delta_j\|_1 \leq O(\epsilon n^2)$, for any $S \subset [n]$ with $|S| \leq n/r$, we have

$$
\sum_{j \in S} \|\Delta_j\|_1 \leq |S| \cdot n2^{l^*} + \sum_{j:\|\Delta_j\|_1 \geq n2^{l^*}} \|\Delta_j\|_1 \leq \epsilon n^2 + O(\epsilon n^2) = O(\epsilon n^2).
$$

The second inequality follows because $|S| \leq n/r$, $2^{l^*} \leq \epsilon r$ and $\sum_{j:\|\Delta_j\|_1 \geq n2^{l^*}} \|\Delta_j\|_1 \leq O(\epsilon n^2)$.
$\square$

### A.5  Proof of Lemma 2.5

*Proof.* Let $M = n^{1/2+1/(2p)}$. Let $Z \in \mathbb{R}^n$ be a random vector where $Z_i$ are i.i.d. random variables with probability density function

$$
g(x) = \begin{cases} f(x)/\Pr[|\Delta_1| \leq M] & \text{if } 0 \leq x \leq M; \\ 0 & \text{otherwise.} \end{cases}
$$

where $f(x)$ is the probability density function of $|\Delta_1|$. Then $\forall a > 0$

$$
\Pr\left[\|\Delta\|_1 \leq a \mid \forall i \in [n], |\Delta_i| \leq M\right] = \Pr\left[\|Z\|_1 \leq a\right].
$$

For $i \in [n]$, because $\mathbf{E}[|\Delta_i|] = 1$, it holds that $\mathbf{E}[Z_i] \leq 1$. We have $\mathbf{E}[\sum_{i=1}^n Z_i] \leq n$. For the second moment, we have

$$
\begin{aligned}
\mathbf{E}[Z_i^2] &= \int_0^M x^2 f(x)/\Pr[|\Delta_1| \leq M]\mathrm{d}x \\
&\leq M^{2-p}/\Pr[|\Delta_1| \leq M] \int_0^M x^p f(x)\mathrm{d}x \\
&\leq O(M^{2-p}) \\
&\leq O(n^{2-p})
\end{aligned}
$$

where the second inequality follows by $\mathbf{E}[|\Delta_1|^p] = O(1)$, and $\Pr[|\Delta_1| \leq M] \geq 1 - \mathbf{E}[|\Delta_1|^p]/M^p \geq 1/2$.

Then by Bernstein's inequality, we have

$$
\begin{aligned}
&\Pr\left[\sum_{i=1}^n Z_i - E\left[\sum_{i=1}^n Z_i\right] \geq \epsilon n\right] \\
&\leq \exp\left(\frac{-0.5\epsilon^2 n^2}{\sum_{i=1}^n \mathbf{E}[Z_i^2] + \frac{1}{3}M \cdot \epsilon n}\right) \\
&\leq e^{-n^{\Theta(1)}}.
\end{aligned}
$$

Thus,

$$
\Pr\left[\|\Delta\|_1 \leq (1+\epsilon)n \mid \forall i \in [n], |\Delta_i| \leq M\right] = \Pr\left[\|Z\|_1 \leq (1+\epsilon)n\right] \geq 1 - e^{-n^{\Theta(1)}}.
$$

$\square$

## A.6 Proof of Lemma 2.10

*Proof.* Recall that $(S_1, S_2, \cdots, S_t, i)$ is equivalent to $(S_{[t]}, i)$. Let $(S_{[t]}, i)$ be an $(A^*, q, t, 1/2)$-good tuple which satisfies $H \cap \left( \bigcup_{j=1}^t S_j \right) = \emptyset$. Let $C$ be the core of $(S_{[t]}, i)$. Let $(x_1, x_2, \cdots, x_t)$ be the coefficients tuple corresponding to $(S_{[t]}, i)$. Then we have that

$$\left\| \frac{1}{|C|} \sum_{j=1}^t A_{S_j} x_j - A_i \right\|_1$$

$$= \left\| \frac{1}{|C|} \sum_{j=1}^t \left( A^*_{S_j} + \Delta_{S_j} \right) x_j - (A^*_i + \Delta_i) \right\|_1$$

$$\leq \left\| \frac{1}{|C|} \sum_{j=1}^t A^*_{S_j} x_j - A^*_i \right\|_1 + \|\Delta_i\|_1 + \frac{1}{|C|} \left\| \sum_{j=1}^t \Delta_{S_j} x_j \right\|_1$$

$$= \|\Delta_i\|_1 + \frac{1}{|C|} \left\| \sum_{j=1}^t \Delta_{S_j} x_j \right\|_1$$

$$\leq \|\Delta_i\|_1 + \frac{2}{t} \left\| \sum_{j=1}^t \Delta_{S_j} x_j \right\|_1$$

$$\leq \|\Delta_i\|_1 + O\left( \frac{1}{t} \cdot (qt)^{1/p} n \right)$$

$$= \|\Delta_i\|_1 + O\left( q^{1/p}/t^{1-1/p} n \right)$$

holds with probability at least $1 - 2^{-n^{\Theta(1)}}$. The first equality follows using $A = A^* + \Delta$. The first inequality follows using the triangle inequality. The second equality follows using the definition of the core and the coefficients tuple (see Definition 2.7 and Definition 2.9). The second inequality follows using Definition 2.7. The third inequality follows by Lemma 2.2 and the condition that $H \cap \left( \bigcup_{j=1}^t S_j \right) = \emptyset$.

Since the size of $\left| \{i\} \cup \left( \bigcup_{j=1}^t S_j \right) \right| = qt + 1$, the total number of $(A^*, q, t, 1/2)-$good tuples is upper bounded by $n^{qt+1} \leq 2^{n^{o(1)}}$. By taking a union bound, we complete the proof. □

## A.7 Proof of Lemma 2.11

*Proof.* For $j \in [t]$, by symmetry of the choices of $S_j$ and $i$, we have $\Pr[i \in R_{A^*}(S_j \cup \{i\})] \leq k/(q+1)$. Thus, by Markov's inequality,

$$\Pr[|\{j \in [t] \mid i \in R_{A^*}(S_j \cup \{i\})\}| > 0.5t]$$
$$\leq \mathbf{E}[|\{j \in [t] \mid i \in R_{A^*}(S_j \cup \{i\})\}|]/(0.5t)$$
$$\leq 2k/q.$$

Thus,

$$\Pr[|\{j \in [t] \mid i \notin R_{A^*}(S_j \cup \{i\})\}| \geq 0.5t] \geq 1 - 2k/q.$$

□

## A.8 Proof of Lemma 2.12

*Proof.* For $S_1, S_2, \cdots, S_t \in \binom{[n]}{q}$ with $\sum_{j=1}^t |S_j| = qt$, define

$$P_{(S_1, S_2, \cdots, S_t)} = \Pr_{i \in [n] \setminus \left( \bigcup_{j=1}^t S_j \right)} [(S_1, S_2, \cdots, S_t, i) \text{ is an } (A^*, q, t, 1/2)-\text{good tuple }].$$

Let set $T$ be defined as follows:

$$\left\{ (S_1, S_2, \cdots, S_t) \,\Big|\, S_1, S_2, \cdots, S_t \in \binom{[n]}{q} \text{ with } \sum_{j=1}^{t} |S_j| = qt \right\}.$$

Let $G$ be the set of all the $(A^*, q, t, 1/2)-$good tuples. Then, we have

$$\Pr_{(S_1, S_2, \cdots, S_t) \sim T} \left[ \left| \left\{ i \in [n] \setminus \left( \cup_{j=1}^{t} S_j \right) \mid (S_1, S_2, \cdots, S_t, i) \in G \right\} \right| \geq (1 - 4k/q)(n - qt) \right]$$

$$= \frac{1}{|T|} \left| \left\{ (S_1, S_2, \cdots, S_t) \mid (S_1, S_2, \cdots, S_t) \in T \text{ and } P_{(S_1, S_2, \cdots, S_t)} \geq 1 - 4k/q \right\} \right|$$

$$= \frac{1}{|T|} \sum_{\substack{(S_1, S_2, \cdots, S_t) \in T \\ P_{(S_1, S_2, \cdots, S_t)} \geq 1 - 4k/q}} 1$$

$$\geq \frac{1}{|T|} \sum_{\substack{(S_1, S_2, \cdots, S_t) \in T \\ P_{(S_1, S_2, \cdots, S_t)} \geq 1 - 4k/q}} P_{(S_1, S_2, \cdots, S_t)}$$

$$\geq 1 - 2k/q - \frac{1}{|T|} \sum_{\substack{(S_1, S_2, \cdots, S_t) \in T \\ P_{(S_1, S_2, \cdots, S_t)} < 1 - 4k/q}} P_{(S_1, S_2, \cdots, S_t)}$$

$$\geq 1 - 2k/q - (1 - 4k/q)$$

$$\geq 2k/q.$$

The second inequality follows from Lemma 2.11

$$\frac{1}{|T|} \sum_{\substack{(S_1, S_2, \cdots, S_t) \in T \\ P_{(S_1, S_2, \cdots, S_t)} < 1 - 4k/q}} P_{(S_1, S_2, \cdots, S_t)} + \frac{1}{|T|} \sum_{\substack{(S_1, S_2, \cdots, S_t) \in T \\ P_{(S_1, S_2, \cdots, S_t)} \geq 1 - 4k/q}} P_{(S_1, S_2, \cdots, S_t)} \geq 1 - 2k/q.$$

$$\square$$

## B Hardness Result

**An overview of the hardness result.** Recall that we overcame the column subset selection lower bound of [24], which shows for entrywise $\ell_1$-low rank approximation that there are matrices for which any subset of $\text{poly}(k)$ columns spans at best a $k^{\Omega(1)}$-approximation. Indeed, we came up with a column subset of size $\text{poly}(k(\epsilon^{-1} + \log n))$ spanning a $(1 + \epsilon)$-approximation. To do this, we assumed $A = A^* + \Delta$, where $A^*$ is an arbitrary rank-$k$ matrix, and the entries are i.i.d. from a distribution with $\mathbf{E}[|\Delta_{i,j}|] = 1$ and $\mathbf{E}[|\Delta_{i,j}|^p] = O(1)$ for any real number $p$ strictly greater than 1.

Here we show an assumption on the moments is necessary, by showing if instead $\Delta$ were drawn from a matrix of i.i.d. Cauchy random variables, for which the $p$-th moment is undefined or infinite for all $p \geq 1$, then for any subset of $n^{o(1)}$ columns, it spans at best a 1.002 approximation. The input matrix $A = n^C 1 \cdot 1^\top + \Delta$, where $C > 0$ is a constant and we show that $n^{\Omega(1)}$ columns need to be chosen to obtain a 1.001-approximation, even for $k = 1$. Note that this result is stronger than that in [24] in that it rules out column subset selection even if one were to choose $n^{o(1)}$ columns; the result in [24] requires at most $\text{poly}(k)$ columns, which for $k = 1$, would just rule out $O(1)$ columns. Our main goal here is to show that a moment assumption on our distribution is necessary, and our result also applies to a symmetric noise distribution which is i.i.d. on all entries, whereas the result of [24] requires a specific deterministic pattern (namely, the identity matrix) on certain entries.

Our main theorem is given in Theorem B.20. The outline of the proof is as follows. We first condition on the event that $\|\Delta\|_1 \leq \frac{4.0002}{\pi} n^2 \ln n$, which is shown in Lemma B.2 and follows form standard analysis of sums of absolute values of Cauchy random variables. Thus, it is sufficient to show if we choose any subset $S$ of $r = n^{o(1)}$ columns, denoted by the submatrix $A_S$, then $\min_{X \in \mathbb{R}^{r \times n}} \|A_S X - A\|_1 \geq \frac{4.01}{\pi} \cdot n^2 \ln n$, as indeed then $\min_{X \in \mathbb{R}^{r \times n}} \|A_S X - A\|_1 \geq 1.002\|\Delta\|_1$

and we rule out a $(1 + \epsilon)$-approximation for $\epsilon$ a sufficiently small constant. To this end, we instead show for a fixed $S$, that $\min_{X \in \mathbb{R}^{r \times n}} \|A_S X - A\|_1 \geq \frac{4.01}{\pi} \cdot n^2 \ln n$ with probability $1 - 2^{-n^{\Theta(1)}}$, and then apply a union bound over all $S$. To prove this for a single subset $S$, we argue that for every "coefficient matrix" $X$, that $\|A_S X - A\|_1 \geq \frac{4.01}{\pi} \cdot n^2 \ln n$.

We show in Lemma B.6, that with probability $1 - (1/n)^{\Theta(n)}$ over $\Delta$, simultaneously for all $X$, if $X$ has a column $X_j$ with $\|X_j\|_1 \geq n^c$ for a constant $c > 0$, then $\|A_S X_j - A_j\|_1 \geq .9n^3$, which is already too large to provide an $O(1)$-approximation. Note that we need such a high probability bound to later union bound over *all* $S$. Lemma B.6 is in turn shown via a net argument on all $X_j$ (it suffices to prove this for a single $j \in [n]$, since there are only $n$ different $j$, so we can union bound over all $j$). The net bounds are given in Definition B.4 and Definition B.5, and the high probability bound for a given coefficient vector $X_j$ is shown in Lemma B.3, where we use properties of the Cauchy distribution. Thus, we can assume $\|X_j\|_1 < n^c$ for all $j \in [n]$. We also show in Fact B.1, conditioned on the fact that $\|\Delta\|_1 \leq \frac{4.002}{\pi} n^2 \ln n$, it holds that for *any* vector $X_j$, if $\|X_j\|_1 < n^c$ and $|1 - \mathbf{1}^\top X_j| > 1 - 10^{-20}$, then $\|A_S X - A\|_1 \geq \|A_S X_j - A_j\|_1 > n^3$. The intuition here is $A = n^{c_0} \mathbf{1} \cdot \mathbf{1}^\top + \Delta$ for a large constant $c_0$, and $X_j$ does not have enough norm ($\|X_j\|_1 \leq n^c$) or correlation with the vector $\mathbf{1}$ ($|1 - \mathbf{1}^\top X_j| > 1 - 10^{-20}$) to make $\|A_S X_j - A_j\|_1$ small.

Given the above, we can assume both that $\|X_j\|_1 \leq n^c$ and $|1 - \mathbf{1}^\top X_j| \leq 1 - 10^{-20}$ for all columns $j$ of our coefficient matrix $X$. We can also assume that $\|A_S X - A\|_1 \leq 4n^2 \ln n$, as otherwise such an $X$ already satisfies $\|A_S X - A\|_1 \geq \frac{4.01}{\pi} \cdot n^2 \ln n$ and we are done. To analyze $\|A_S X - A\|_1 = \sum_{i,j} |(A_S X - A_{[n]\setminus S})_{i,j}|$ in Theorem B.20, we then split the sum over "large coordinates" $(i,j)$ for which $|\Delta_{i,j}| > n^{1.0002}$, and "small coordinates" $(i,j)$ for which $|\Delta_{i,j}| < n^{.9999}$, and since we seek to lower bound $\|A_S X - A_{[n]\setminus S}\|_1$, we drop the remaining coordinates $(i,j)$. To handle large coordinates, we observe that since the column span of $A_S$ is only $r = n^{o(1)}$-dimensional, as one ranges over all vectors $y$ in its span of 1-norm, say, $O(n^2 \ln n)$, there is only a small subset $T$, of size at most $n^{.99999}$ of coordinates $i \in [n]$ for which we could ever have $|y_i| \geq n^{1.0001}$. We show this in Lemma B.9. This uses the property of vectors in low-dimensional subspaces, and has been exploited in earlier works in the context of designing so-called subspace embeddings [2, 3]. We call $T$ the "bad region" for $A_S$. While the column span of $A_S$ depends on $\Delta_S$, it is independent of $\Delta_{[n]\setminus S}$, and thus it is extremely unlikely that the large coordinate of $\Delta_S$ "match up" with the bad region of $A_S$. This is captured in Lemma B.13, where we show that if $\|A_S X - A_{[n]\setminus S}\|_1 \leq 4n^2 \ln n$ (as we said we could assume above), then $\sum_{\text{large coordinates } i,j} |(A_S X - A_{[n]\setminus S})_{i,j}|$ is at least $\frac{1.996}{\pi} n^2 \ln n$. Intuitively, the heavy coordinates make up about $\frac{2}{\pi} n^2 \ln n$ of the total mass of $\|\Delta\|_1$, by tail bounds of the Cauchy distribution, and for any set $S$ of size $n^{o(1)}$, $A_S$ fits at most a small portion of this, still leaving us left with $\frac{1.996}{\pi} n^2 \ln n$ in cost. Our goal is to show that $\|A_S X - A_{[n]\setminus S}\|_1 \geq \frac{4.01}{\pi} \cdot n^2 \ln n$, so we still have a way to go.

We next analyze $\sum_{\text{small coordinates } i,j} |(A_S X - A_{[n]\setminus S})_{i,j}|$. Via Bernstein's inequality, in Lemma B.14 we argue that for any fixed vector $y$ and random vector $\Delta_j$ of i.i.d. Cauchy entries, roughly half of the contribution of coordinates to $\|\Delta_j\|_1$ will come from coordinates $j$ for which $\text{sign}(y_j) = \text{sign}(\Delta_j)$ and $|\Delta_j| \leq n^{.9999}$, giving us a contribution of roughly $\frac{.9998}{\pi} n \ln n$ to the cost. The situation we will actually be in, when analyzing a column of $A_S X - A_{[n]\setminus S}$, is that of taking the sum of two independent Cauchy vectors, shifted by a multiple of $\mathbf{1}^\top$. We analyze this setting in Lemma B.16, after first conditioning on certain level sets having typical behavior in Lemma B.15. This roughly doubles the contribution, gives us roughly a contribution of $\frac{1.996}{\pi} n^2 \ln n$ from coordinates $j$ for which $(i,j)$ is a small coordinate and we look at coordinates $i$ on which the sum of two independent Cauchy vectors have the same sign. Combined with the contribution from the heavy coordinates, this gives us a cost of roughly $\frac{3.992}{\pi} n^2 \ln n$, which still falls short of the $\frac{4.01}{\pi} \cdot n^2 \ln n$ total cost we are aiming for. Finally, if we sum up two independent Cauchy vectors and look at the contribution to the sum from coordinates which disagree in sign, due to the anti-concentration of the Cauchy distribution we can still "gain a little bit of cost" since the values, although differing in sign, are still likely not to be very close in magnitude. We formalize this in Lemma B.17. We combine all of the costs from small coordinates in Lemma B.18, where we show we obtain a contribution of at least $\frac{2.025}{\pi} n \ln n$. This is enough, when combined with our earlier $\frac{1.996}{\pi} n^2 \ln n$ contribution from the heavy coordinates, to obtain an overall $\frac{4.01}{\pi} \cdot n^2 \ln n$ lower bound on the cost, and conclude the proof of our main theorem in Theorem B.20.

In the remaining sections, we will present detailed proofs.

## B.1 A Useful Fact

**Fact B.1.** *Let $c_0 > 0$ be a sufficiently large constant. Let $u = n^{c_0} \cdot \mathbf{1} \in \mathbb{R}^n$ and $\Delta \in \mathbb{R}^{n \times (d+1)}$. If $\sum_{i=1}^{d+1} \|\Delta_i\|_1 \leq n^3$ and if $\alpha \in \mathbb{R}^d$ satisfies $|1 - \mathbf{1}^\top \alpha| > 1/n^{c_1}$ and $\|\alpha\|_1 \leq n^c$, where $0 < c < c_0 - 10$ is a constant and $c_1 > 3$ is another constant depending on $c_0, c$, then*

$$\|u - u\mathbf{1}^\top \alpha + \Delta_{d+1} - \Delta_{[d]}\alpha\|_1 > n^3.$$

*Proof.*

$$\|u - u\mathbf{1}^\top \alpha + \Delta_{d+1} - \Delta_{[d]}\alpha\|_1$$
$$\geq |1 - \mathbf{1}^\top \alpha| \cdot \|u\|_1 - \|\Delta_{d+1}\|_1 - \|\Delta_{[d]}\alpha\|_1$$
$$\geq |1 - \mathbf{1}^\top \alpha| \cdot n \cdot n^{c_0} - n^3 - n^4\|\alpha\|_1$$
$$\geq |1 - \mathbf{1}^\top \alpha| \cdot n \cdot n^{c_0} - n^{5+c}$$
$$\geq n^{c_0+1-c_1} - n^{5+c}$$
$$\geq n^3.$$

The first inequality follows by the triangle inequality. The second inequality follows since $u = n^{c_0} \cdot \mathbf{1} \in \mathbb{R}^n$ and $\sum_{i=1}^{d+1} \|\Delta_i\|_1 \leq n^3$. The third inequality follows since $\|\alpha\|_1 \leq n^c$. The fourth inequality follows since $|1 - \mathbf{1}^\top \alpha| > 1/n^{c_1}$. The last inequality follows since $c_0 - c_1 > c + 5$. $\square$

## B.2 One-Sided Error Concentration Bound for a Random Cauchy Matrix

**Lemma B.2** (Lower bound on the cost). *If $n$ is sufficiently large, then*

$$\Pr_{\Delta \sim \{C(0,1)\}^{n \times n}} \left[ \|\Delta\|_1 \leq \frac{4.0002}{\pi} n^2 \ln n \right] \geq 1 - O(1/\log\log n).$$

*Proof.* Let $\Delta \in \mathbb{R}^{n \times n}$ be a random matrix such that each entry is an i.i.d. $C(0,1)$ random Cauchy variable. Let $B = n^2 \ln \ln n$. Let $Z \in \mathbb{R}^{n \times n}$ and $\forall i, j \in [n]$,

$$Z_{i,j} = \begin{cases} |\Delta_{i,j}| & |\Delta_{i,j}| < B \\ B & \text{Otherwise} \end{cases}.$$

For fixed $i, j \in [n]$, we have

$$\mathbf{E}[Z_{i,j}] = \frac{2}{\pi} \int_0^B \frac{x}{1+x^2} \mathrm{d}x + \Pr[|\Delta_{i,j}| \geq B] \cdot B$$
$$= \frac{1}{\pi} \ln(B^2 + 1) + \Pr[|\Delta_{i,j}| \geq B] \cdot B$$
$$\leq \frac{1}{\pi} \ln(B^2 + 1) + 1$$

where the first inequality follows by the cumulative distribution function of a half Cauchy random variable. We also have $\mathbf{E}[Z_{i,j}] \geq \frac{1}{\pi} \ln(B^2 + 1)$. For the second moment, we have

$$\mathbf{E}[Z_{i,j}^2] = \frac{2}{\pi} \int_0^B \frac{x^2}{1+x^2} \mathrm{d}x + \Pr[|\Delta_{i,j}| \geq B] \cdot B^2$$
$$= \frac{2}{\pi}(B - \tan^{-1} B) + \Pr[|\Delta_{i,j}| \geq B] \cdot B^2$$
$$\leq \frac{2}{\pi} B + B$$
$$\leq 2B$$

where the first inequality follows by the cumulative distribution function of a half Cauchy random variable. By applying Bernstein's inequality, we have

$$\Pr\left[\|Z\|_1 - \mathbf{E}[\|Z\|_1] > 0.0001\,\mathbf{E}[\|Z\|_1]\right]$$

$$\leq \exp\left(-\frac{0.5 \cdot 0.0001^2\,\mathbf{E}[\|Z\|_1]^2}{n^2 \cdot 2B + \frac{1}{3}B \cdot 0.0001\,\mathbf{E}[\|Z\|_1]}\right)$$

$$\leq \exp(-\Omega(\ln n / \ln\ln n))$$

$$\leq O(1/\ln n). \tag{6}$$

The first inequality follows by the definition of $Z$ and the second moment of $Z_{i,j}$. The second inequality follows from $\mathbf{E}[\|Z\|_1] = \Theta(n^2 \ln n)$ and $B = \Theta(n^2 \ln\ln n)$. Notice that

$$\Pr\left[\|\Delta\|_1 > \frac{4.0002}{\pi}n^2\ln n\right]$$

$$= \Pr\left[\|\Delta\|_1 > \frac{4.0002}{\pi}n^2\ln n \mid \forall i,j, |\Delta_{i,j}| < B\right]\Pr\left[\forall i,j, |\Delta_{i,j}| < B\right]$$

$$+ \Pr\left[\|\Delta\|_1 > \frac{4.0002}{\pi}n^2\ln n \mid \exists i,j, |\Delta_{i,j}| \geq B\right]\Pr\left[\exists i,j, |\Delta_{i,j}| \geq B\right]$$

$$\leq \Pr\left[\|\Delta\|_1 > \frac{4.0002}{\pi}n^2\ln n \mid \forall i,j, |\Delta_{i,j}| < B\right] + \Pr\left[\exists i,j, |\Delta_{i,j}| \geq B\right]$$

$$\leq \Pr\left[\|Z\|_1 > \frac{4.0002}{\pi}n^2\ln n\right] + \Pr\left[\exists i,j, |\Delta_{i,j}| \geq B\right]$$

$$\leq \Pr\left[\|Z\|_1 > \frac{4.0002}{\pi}n^2\ln n\right] + n^2 \cdot 1/B$$

$$\leq \Pr\left[\|Z\|_1 > 1.0001\,\mathbf{E}[\|Z\|_1]\right] + n^2 \cdot 1/B$$

$$\leq O(1/\log(n)) + O(1/\log\log n)$$

$$\leq O(1/\log\log n)$$

The second inequality follows by the definition of $Z$. The third inequality follows by the union bound and the cumulative distribution function of a half Cauchy random variable. The fourth inequality follows from $\mathbf{E}[\|Z\|_1] \leq n^2(1/\pi \cdot \ln(B^2 + 1) + 1) \leq 4.0000001/\pi \cdot n^2 \ln n$ when $n$ is sufficiently large. $\qquad\square$

### B.3  "For Each" Guarantee

In the following Lemma, we show that, for each fixed coefficient vector $\alpha$, if the entry of $\alpha$ is too large, the fitting cost cannot be small.

**Lemma B.3** (For each fixed $\alpha$, the entry cannot be too large). *Let $c > 0$ be a sufficiently large constant, $n \geq d \geq 1$, $u \in \mathbb{R}^n$ be any fixed vector and $\Delta \in \mathbb{R}^{n \times d}$ be a random matrix where $\forall i \in [n], j \in [d], \Delta_{i,j} \sim C(0,1)$ independently. For any fixed $\alpha \in \mathbb{R}^d$ with $\|\alpha\|_1 = n^c$,*

$$\Pr_{\Delta \sim \{C(0,1)\}^{n \times d}}[\|(u \cdot \mathbf{1}^\top + \Delta)\alpha\|_1 > n^3] > 1 - (1/n)^{\Theta(n)}.$$

*Proof.* Let $c$ be a sufficiently large constant. Let $\alpha \in \mathbb{R}^d$ with $\|\alpha\|_1 = n^c$. Let $u \in \mathbb{R}^n$ be any fixed vector. Let $\Delta \in \mathbb{R}^{n \times d}$ be a random matrix where $\forall i \in [n], j \in [d], \Delta_{i,j} \sim C(0,1)$. Then $\Delta\alpha \in \mathbb{R}^n$ is a random vector with each entry drawn independently from $C(0, \|\alpha\|_1)$. Due to the probability density function of standard Cauchy random variables,

$$\Pr[\|\Delta\alpha\|_1 < n^3] \geq \Pr[\|\Delta\alpha + u \cdot \mathbf{1}^\top\alpha\|_1 < n^3].$$

It suffices to upper bound $\Pr[\|\Delta\alpha\|_1 < n^3]$. If $c > 10$, then due to the cumulative distribution function of Cauchy random variables, for a fixed $i \in [n]$, $\Pr[|(\Delta\alpha)_i| < n^3] < 1/n$. Thus, $\Pr[\|\Delta\alpha\|_1 < n^3] < (\frac{1}{n})^n$. Thus,

$$\Pr_{\Delta \sim \{C(0,1)\}^{n \times d}}[\|(u \cdot \mathbf{1}^\top + \Delta)\alpha\|_1 > n^3] > 1 - (1/n)^n.$$

$\qquad\square$

## B.4 From "For Each" to "For All" via an $\epsilon$-Net

**Definition B.4** ($\epsilon$-net for the $\ell_1$-norm ball). *Let $A \in \mathbb{R}^{n \times d}$ have rank $d$, and let $L = \{y \in \mathbb{R}^n \mid y = Ax, x \in \mathbb{R}^d\}$ be the column space of $A$. An $\epsilon$-net of the $\ell_1$-unit sphere $\mathcal{S}^{d-1} = \{y \mid \|y\|_1 = 1, y \in L\} \subset L$ is a set $N \subset \mathcal{S}^{d-1}$ of points for which $\forall y \in \mathcal{S}^{d-1}, \exists y' \in N$ for which $\|y - y'\| \leq \epsilon$.*

[33] proved an upper bound on the size of an $\epsilon$-net.

**Lemma B.5** (See, e.g., the ball $B$ on page 2068 of [33]). *Let $A \in \mathbb{R}^{n \times d}$ have rank $d$, and let $L = \{y \in \mathbb{R}^n \mid y = Ax, x \in \mathbb{R}^d\}$ be the column space of $A$. For $\epsilon \in (0, 1)$, an $\epsilon$-net (Definition B.4) $N$ of the $\ell_1$-unit sphere $\mathcal{S}^{d-1} = \{y \mid \|y\|_1 = 1, y \in L\} \subset L$ exists. Furthermore, the size of $N$ is at most $(3/\epsilon)^d$.*

**Lemma B.6** (For all possible $\alpha$, the entry cannot be too large). *Let $n \geq 1, d = n^{o(1)}$. Let $u = n^{c_0} \cdot \mathbf{1} \in \mathbb{R}^n$ denote a fixed vector where $c_0$ is a constant. Let $\Delta \in \mathbb{R}^{n \times d}$ be a random matrix where $\forall i \in [n], j \in [d], \Delta_{i,j} \sim C(0, 1)$ independently. Let $c > 0$ be a sufficiently large constant. Conditioned on $\|\Delta\|_1 \leq n^3$, with probability at least $1 - (1/n)^{\Theta(n)}$, for all $\alpha \in \mathbb{R}^d$ with $\|\alpha\|_1 \geq n^c$, we have $\|(u \cdot \mathbf{1}^\top + \Delta)\alpha\|_1 > 0.9n^3$.*

*Proof.* Due to Lemma B.5, there is a set $N \subset \{\alpha \in \mathbb{R}^d \mid \|\alpha\|_1 = n^c\} \subset \mathbb{R}^d$ with $|N| \leq 2^{\Theta(d \log n)}$ such that $\forall \alpha \in \mathbb{R}^d$ with $\|\alpha\|_1 = n^c, \exists \alpha' \in N$ such that $\|\alpha - \alpha'\|_1 \leq 1/n^{c'}$ where $c' > c_0 + 100$ is a constant. By applying Lemma B.3 and union bounding over all the points in $N$, with probability at least $1 - (1/n)^n \cdot |N| \geq 1 - (1/n)^n \cdot 2^{n^{o(1)}} = 1 - (1/n)^{\Theta(n)}, \forall \alpha' \in N, \|(u \cdot \mathbf{1}^\top + \Delta)\alpha'\|_1 > n^3$. $\forall \alpha \in \mathbb{R}^d$ with $\|\alpha\|_1 = n^c$, we can find $\alpha' \in N$ such that $\|\alpha - \alpha'\|_1 \leq 1/n^{c'}$. Let $\gamma = \alpha - \alpha'$. Then,

$$
\begin{aligned}
&\|(u \cdot \mathbf{1}^\top + \Delta)\alpha\|_1 \\
&= \|(u \cdot \mathbf{1}^\top + \Delta)(\alpha' + \gamma)\|_1 \\
&\geq \|(u \cdot \mathbf{1}^\top + \Delta)\alpha'\|_1 - \|(u \cdot \mathbf{1}^\top + \Delta)\gamma\|_1 \\
&\geq n^3 - \sqrt{n}\|(u \cdot \mathbf{1}^\top + \Delta)\gamma\|_2 \\
&\geq n^3 - \sqrt{n}(\|u \cdot \mathbf{1}^\top\|_2 + \|\Delta\|_2)\|\gamma\|_2 \\
&\geq n^3 - n^{c_0+50}/n^{c'} \\
&\geq 0.9n^3.
\end{aligned}
$$

The first equality follows from $\alpha = \alpha' + \gamma$. The first inequality follows by the triangle inequality. The second inequality follows by the relaxation from the $\ell_1$ norm to the $\ell_2$ norm. The third inequality follows from the operator norm and the triangle inequality. The fourth inequality follows using $\|\Delta\|_2 \leq \|\Delta\|_1 \leq n^3, \|u\|_2 \leq n^{c_0+10}, \|\gamma\|_2 \leq \|\gamma\|_1 \leq (1/n)^{c'}$. The last inequality follows since $c' > c_0 + 100$.

For $\alpha \in \mathbb{R}^n$ with $\|\alpha\|_1 > n^c$, let $\alpha' = \alpha/\|\alpha\|_1 \cdot n^c$. Then

$$\|(u \cdot \mathbf{1}^\top + \Delta)\alpha\|_1 \geq \|(u \cdot \mathbf{1}^\top + \Delta)\alpha'\|_1 \geq 0.9n^3.$$

$\square$

## B.5 Bounding the Cost from the Large-Entry Part via "Bad" Regions

In this section, we will use the concept of *well-conditioned basis* in our analysis.

**Definition B.7** (Well-conditioned basis [33]). *Let $A \in \mathbb{R}^{n \times m}$ have rank $d$. Let $p \in [1, \infty)$, and let $\| \cdot \|_q$ be the dual norm of $\| \cdot \|_p$, i.e., $1/p + 1/q = 1$. If $U \in \mathbb{R}^{n \times d}$ satisfies*

    *1. $\|U\|_p \leq \alpha$,*

    *2. $\forall z \in \mathbb{R}^d, \|z\|_q \leq \beta\|Uz\|_p$,*

*then $U$ is an $(\alpha, \beta, p)$ well-conditioned basis for the column space of $A$.*

The following theorem gives an existence result of a well-conditioned basis.

**Theorem B.8** ($\ell_1$ well-conditioned basis [33])**.** *Let $A \in \mathbb{R}^{n \times m}$ have rank d. There exists $U \in \mathbb{R}^{n \times d}$ such that U is a $(d, 1, 1)$ well-conditioned basis for the column space of A.*

In the following lemma, we consider vectors from low-dimensional subspaces. For a coordinate, if there is a vector from the subspace for which this entry is large, but the norm of the vector is small, then this kind of coordinate is pretty "rare". More formally,

**Lemma B.9.** *Given a matrix $U \in \mathbb{R}^{n \times r}$ for a sufficiently large $n \geq 1$, let $r = n^{o(1)}$. Let $S = \{y | y = Ux, x \in \mathbb{R}^r\}$. Let the set T denote $\{i \in [n] \mid \exists y \in S, |y_i| \geq n^{1.0001}$ and $\|y\|_1 < 8n^2 \ln n\}$. Then we have*

$$|T| \leq n^{0.99999}.$$

*Proof.* Due to Theorem B.8, let $U \in \mathbb{R}^{n \times r}$ be the $(r, 1, 1)$ well-conditioned basis of the column space of U. If $i \in T$, then $\exists x \in \mathbb{R}^r$ such that $|(Ux)_i| \geq n^{1.0001}$ and $\|Ux\|_1 < 8n^2 \ln n$. Thus, we have

$$n^{1.0001} \leq |(Ux)_i| \leq \|U^i\|_1 \|x\|_\infty \leq \|U^i\|_1 \|Ux\|_1 \leq \|U^i\|_1 \cdot 8n^2 \ln n.$$

The first inequality follows using $n^{1.0001} \leq |(Ux)_i|$. The second inequality follows by Hölder's inequality. The third inequality follows by the second property of the well-conditioned basis. The fourth inequality follows using $\|Ux\|_1 < 8n^2 \ln n$. Thus, we have

$$\|U^i\|_1 \geq n^{1.0001}/n^{2+o(1)} \geq 1/n^{0.9999-o(1)}.$$

Notice that $\sum_{j=1}^{n} \|U^j\|_1 = \|U\|_1 \leq r$. Thus,

$$|T| \leq r/(1/n^{0.9999-o(1)}) = n^{0.9999+o(1)} \leq n^{0.99999}.$$

$\square$

**Definition B.10** (Bad region)**.** *Given a matrix $U \in \mathbb{R}^{n \times r}$, we say $\mathcal{B}(U) = \{i \in [n] \mid \exists y \in \mathrm{colspan}(U) \subset \mathbb{R}^n$ s.t. $y_i \geq n^{1.0001}$ and $\|y\|_1 \leq 8n^2 \ln n\}$ is a bad region for U.*

Next we state a lower and an upper bound on the probability that a Cauchy random variable is in a certain range,

**Claim B.11.** *Let $X \sim C(0, 1)$ be a standard Cauchy random variable. Then for any $x > 1549$,*

$$\frac{2}{\pi} \cdot \frac{\ln(1.001)}{x} \geq \Pr[|X| \in (x, 1.001x]] \geq \frac{1.999}{\pi} \cdot \frac{\ln(1.001)}{x}.$$

*Proof.* When $x > 1549$, $\frac{2}{\pi} \cdot \frac{\ln(1.001)}{x} \geq \frac{2}{\pi} \cdot (\tan^{-1}(1.001x) - \tan^{-1}(x)) \geq \frac{1.999}{\pi} \cdot \frac{\ln(1.001)}{x}.$ $\square$

We build a level set for the "large" noise values, and we show the bad region cannot cover much of the large noise. The reason is that the bad region is small, and for each row, there is always some large noise.

**Lemma B.12.** *Given a matrix $U \in \mathbb{R}^{n \times r}$ with n sufficiently large, let $r = n^{o(1)}$, and consider a random matrix $\Delta \in \mathbb{R}^{n \times (n-r)}$ with $\Delta_{i,j} \sim C(0, 1)$ independently. Let $L_t = \{(i, j) \mid (i, j) \in [n] \times [n - r], |\Delta_{i,j}| \in (1.001^t, 1.001^{t+1}]\}$. With probability at least $1 - 1/2^{n^{\Theta(1)}}$, for all $t \in (\frac{1.0002 \ln n}{\ln 1.001}, \frac{1.9999 \ln n}{\ln 1.001}) \cap \mathbb{N}$,*

$$|L_t \setminus (\mathcal{B}(U) \times [n - r])| \geq n(n - r) \cdot 1.998 \cdot \ln(1.001)/(\pi \cdot 1.001^t).$$

*Proof.* Let $N = n \cdot (n - r)$. Then according to Claim B.11, $\forall t \in (\frac{1.0002 \ln n}{\ln 1.001}, \frac{1.9999 \ln n}{\ln 1.001}) \cap \mathbb{N}$, $\mathbf{E}(|L_t|) \geq N \cdot 1.999 \cdot \ln(1.001)/(\pi \cdot 1.001^t) \geq n^{\Theta(1)}$. For a fixed $t$, by a Chernoff bound, with probability at least $1 - 1/2^{n^{\Theta(1)}}$, $|L_t| \geq N \cdot 1.9989 \cdot \ln(1.001)/(\pi \cdot 1.001^t)$. Due to Lemma B.9, $|\mathcal{B}(U) \times [n - r]| \leq n^{0.99999}(n - r) = N/n^{0.00001}$. Due to the Chernoff bound, with probability at least $1 - 1/2^{n^{\Theta(1)}}$, $|L_t \cap (\mathcal{B}(U) \times [n - r])| < N/n^{0.00001} \cdot 2.0001 \cdot \ln(1.001)/(\pi \cdot 1.001^t)$. Thus, with probability at least $1 - 1/2^{n^{\Theta(1)}}$, $|L_t \setminus (\mathcal{B}(U) \times [n - r])| \geq N \cdot 1.998 \cdot \ln(1.001)/(\pi \cdot 1.001^t)$. By taking a union bound over all $t \in (\frac{1.0002 \ln n}{\ln 1.001}, \frac{1.9999 \ln n}{\ln 1.001}) \cap \mathbb{N}$, we complete the proof. $\square$

**Lemma B.13** (The cost of the large noise part)**.** *Let $n \geq 1$ be sufficiently large, and let $r = n^{o(1)}$. Given a matrix $U \in \mathbb{R}^{n \times r}$, and a random matrix $\Delta \in \mathbb{R}^{n \times (n-r)}$ with $\Delta_{i,j} \sim C(0,1)$ independently, let $\mathcal{I} = \{(i,j) \in [n] \times [n-r] \mid |\Delta_{i,j}| \geq n^{1.0002}\}$. If $\|\Delta\|_1 \leq 4n^2 \ln n$, then with probability at least $1 - 1/2^{n^{\Theta(1)}}$, for all $X \in \mathbb{R}^{r \times n}$, either*

$$\sum_{(i,j) \in \mathcal{I}} |(UX - \Delta)_{i,j}| > \frac{1.996}{\pi} n^2 \ln n,$$

*or*

$$\|UX - \Delta\|_1 > 4n^2 \ln n$$

*Proof.*

$$
\begin{aligned}
&\sum_{(i,j) \in \mathcal{I}} |(UX - \Delta)_{i,j}| \\
&\geq \sum_{(i,j) \in \mathcal{I} \setminus \mathcal{B}(U)} |(UX - \Delta)_{i,j}| \\
&\geq \sum_{(i,j) \in \mathcal{I} \setminus \mathcal{B}(U)} |(\Delta)_{i,j}| - \sum_{(i,j) \in \mathcal{I} \setminus \mathcal{B}(U)} |(UX)_{i,j}| \qquad (7)
\end{aligned}
$$

Let $N = n(n - r)$. By a Chernoff bound and the cumulative distribution function of a Cauchy random variable, with probability at least $1 - 1/2^{n^{\Theta(1)}}$, $|\mathcal{I}| \leq 1.1 \cdot N/n^{1.0002}$. If $\exists (i,j) \in \mathcal{I} \setminus \mathcal{B}(U)$ which has $|(UX)_{i,j}| > n^{1.0001}$, then according to the definition of $\mathcal{B}(U)$, $\|UX\|_1 \geq \|(UX)_j\|_1 \geq 8n^2 \ln n$. Due to the triangle inequality, $\|UX - \Delta\|_1 \geq \|UX\|_1 - \|\Delta\|_1 \geq 4n^2 \ln n$. If $\forall (i,j) \in \mathcal{I} \setminus \mathcal{B}(U)$ we have $|(UX)_{i,j}| \leq n^{1.0001}$, then

$$\sum_{(i,j) \in \mathcal{I} \setminus \mathcal{B}(U)} |(UX)_{i,j}| \leq |\mathcal{I}| \cdot n^{1.0001} \leq 1.1 \cdot N/n^{0.0001}. \qquad (8)$$

Due to Lemma B.12, with probability at least $1 - 1/2^{n^{\Theta(1)}}$,

$$
\begin{aligned}
&\sum_{(i,j) \in \mathcal{I} \setminus \mathcal{B}(U)} |(\Delta)_{i,j}| \\
&\geq \sum_{t \in (\frac{1.0002 \ln n}{\ln 1.001}, \frac{1.9999 \ln n}{\ln 1.001}) \cap \mathbb{N}} 1.001^t \cdot N \cdot 1.998 \cdot \ln(1.001)/(\pi \cdot 1.001^t) \\
&\geq \frac{1.997}{\pi} \cdot N \ln n. \qquad (9)
\end{aligned}
$$

We plug (8) and (9) into (7), from which we have

$$\sum_{(i,j) \in \mathcal{I}} |(UX - \Delta)_{i,j}| \geq \frac{1.996}{\pi} n^2 \ln n.$$

$\square$

## B.6   Cost from the Sign-Agreement Part of the Small-Entry Part

We use $-y$ to fit $\Delta$ (we think of $A_S \alpha = A_S^* \alpha - y$, and want to minimize $\| - y - \Delta\|_1$). If the sign of $y_j$ is the same as the sign of $\Delta_j$, then both coordinate values will collectively contribute.

**Lemma B.14** (The contribution from $\Delta_i$ when $\Delta_i$ and $y_i$ have the same sign)**.** *Suppose we are given a vector $y \in \mathbb{R}^n$ and a random vector $\Delta \in \mathbb{R}^n$ with $\Delta_j \sim C(0,1)$ independently. Then with probability at least $1 - 1/2^{n^{\Theta(1)}}$,*

$$\sum_{j \,:\, \text{sign}(y_j) = \text{sign}(\Delta_j) \text{ and } |\Delta_j| \leq n^{0.9999}} |\Delta_j| > \frac{0.9998}{\pi} n \ln n.$$

*Proof.* For $j \in [n]$, define the random variable

$$Z_j = \begin{cases} \Delta_j & 0 < \Delta_j \leq n^{0.9999} \\ 0 & \text{otherwise} \end{cases}.$$

Then, we have

$$\Pr\left[\sum_{j \,:\, \text{sign}(y_j)=\text{sign}(\Delta_{i,j}) \text{ and } |\Delta_j|\leq n^{0.9999}} |\Delta_j| > \frac{0.9998}{\pi} n \ln n\right] = \Pr\left[\sum_{j=1}^{n} Z_j > \frac{0.9998}{\pi} n \ln n\right].$$

Let $B = n^{0.9999}$. For $j \in [n]$,

$$\mathbf{E}[Z_j] = \frac{1}{\pi}\int_0^B \frac{x}{1+x^2}\mathrm{d}x = \frac{1}{2\pi}\ln(B^2+1).$$

Also,

$$\mathbf{E}[Z_j^2] = \frac{1}{\pi}\int_0^B \frac{x^2}{1+x^2}\mathrm{d}x = \frac{B - \tan^{-1}(B)}{\pi} \leq B.$$

By Bernstein's inequality,

$$\Pr\left[\mathbf{E}\left[\sum_{j=1}^{n} Z_j\right] - \sum_{j=1}^{n} Z_j > 10^{-5}\,\mathbf{E}\left[\sum_{j=1}^{n} Z_j\right]\right]$$

$$\leq \exp\left(-\frac{0.5 \cdot \left(10^{-5}\,\mathbf{E}\left[\sum_{j=1}^{n} Z_j\right]\right)^2}{\sum_{j=1}^{n}\mathbf{E}[Z_j^2] + \frac{1}{3}B \cdot 10^{-5}\,\mathbf{E}\left[\sum_{j=1}^{n} Z_j\right]}\right)$$

$$\leq \exp\left(-\frac{5 \cdot 10^{-11}n^2\ln^2(B^2+1)/(4\pi^2)}{nB + \frac{1}{3}B \cdot 10^{-5}n\ln(B^2+1)/(2\pi)}\right)$$

$$\leq e^{-n^{\Theta(1)}}.$$

The last inequality follows since $B = n^{0.9999}$. Thus, we have

$$\Pr\left[\sum_{j=1}^{n} Z_j < 0.9998/\pi \cdot n \ln n\right] \leq \Pr\left[\sum_{j=1}^{n} Z_j < 0.99999n\ln(B^2+1)/(2\pi)\right] \leq e^{-n^{\Theta(1)}}.$$

$\square$

**Lemma B.15** (Bound on level sets of a Cauchy vector). *Suppose we are given a random vector $y \in \mathbb{R}^n$ with $y_i \sim C(0,1)$ chosen independently. Let*

$$L_t^- = \{i \in [n] \mid -y_i \in (1.001^t, 1.001^{t+1}]\} \text{ and } L_t^+ = \{i \in [n] \mid y_i \in (1.001^t, 1.001^{t+1}]\}.$$

*With probability at least $1 - 1/2^{n^{\Theta(1)}}$, for all $t \in (\frac{\ln 1549}{\ln 1.001}, \frac{0.9999 \ln n}{\ln 1.001}) \cap \mathbb{N}$,*

$$\min(|L_t^-|, |L_t^+|) \geq 0.999n \cdot \frac{1}{\pi}\frac{\ln 1.001}{1.001^t}.$$

*Proof.* For $i \in [n], t \geq \frac{\ln 1549}{\ln 1.001}$, according to Claim B.11, $\Pr[y_i \in (1.001^t, 1.001^{t+1}]] \geq 0.9995/\pi \cdot \ln(1.0001)/1.001^t$. Thus, $\mathbf{E}[|L_t^+|] = \mathbf{E}[|L_t^-|] = n \cdot 0.9995/\pi \cdot \ln(1.0001)/1.001^t$. Since $t \leq \frac{0.9999 \ln n}{\ln 1.001}$, $1.001^t \leq n^{0.9999}$, we have $\mathbf{E}[|L_t^+|] = \mathbf{E}[|L_t^-|] \geq n^{\Theta(1)}$. By applying a Chernoff bound,

$$\Pr[|L_t^+| > 0.999n/\pi \cdot \ln(1.0001)/1.001^t] \geq 1 - 1/2^{n^{\Theta(1)}}.$$

Similarly, we have

$$\Pr[|L_t^-| > 0.999n/\pi \cdot \ln(1.0001)/1.001^t] \geq 1 - 1/2^{n^{\Theta(1)}}.$$

By taking a union bound over all the $L_t^+$ and $L_t^-$, we complete the proof. $\square$

**Lemma B.16** (The contribution from $y_i$ when $\Delta_i$ and $y_i$ have the same sign). *Let $u = \eta \cdot \mathbf{1} \in \mathbb{R}^n$ where $\eta \in \mathbb{R}$ is an arbitrary real number. Let $y \in \mathbb{R}^n$ be a random vector with $y_i \sim C(0, \beta)$ independently for some $\beta > 0$. Let $\Delta \in \mathbb{R}^n$ be a random vector with $\Delta_i \sim C(0, 1)$ independently. With probability at least $1 - 1/2^{n^{\Theta(1)}}$,*

$$\sum_{i\,:\,\mathrm{sign}((u+y)_i)=\mathrm{sign}(\Delta_i)\text{ and }|\Delta_i|\leq n^{0.9999}} |(u+y)_i| \geq \beta \cdot \frac{0.997}{\pi} n \ln n.$$

*Proof.* For all $t \in (\frac{\ln 1549}{\ln 1.001}, \frac{0.9999 \ln n}{\ln 1.001}) \cap \mathbb{N}$, define
$L_t^- = \{i \in [n] \mid -y_i \in (\beta \cdot 1.001^t, \beta \cdot 1.001^{t+1}]\}$ and $L_t^+ = \{i \in [n] \mid y_i \in (\beta \cdot 1.001^t, \beta \cdot 1.001^{t+1}]\}$.
Define

$$G = \{i \in [n] \mid \mathrm{sign}((u+y)_i) = \mathrm{sign}(\Delta_i) \text{ and } |\Delta_i| \leq n^{0.9999}\}.$$

Then $\forall i \in [n], \Pr[i \in G] \geq 0.5 - 1/n^{0.9999} \geq 0.4999999999$. Due to Lemma B.15,

$$\min(|L_t^-|, |L_t^+|) \geq 0.999n \cdot \frac{1}{\pi} \frac{\ln 1.001}{1.001^t} \geq n^{\Theta(1)}.$$

By a Chernoff bound and a union bound, with probability at least $1 - 1/2^{n^{\Theta(1)}}$, $\forall t \in (\frac{\ln 1549}{\ln 1.001}, \frac{0.9999 \ln n}{\ln 1.001}) \cap \mathbb{N}$,

$$\min(|L_t^- \cap G|, |L_t^+ \cap G|)$$
$$\geq 0.499n \cdot \frac{1}{\pi} \frac{\ln 1.001}{1.001^t}. \tag{10}$$

Then we have

$$\sum_{i \in G} |(u+y)_i|$$

$$\geq \sum_{t \in (\frac{\ln 1549}{\ln 1.001}, \frac{0.9999 \ln n}{\ln 1.001}) \cap \mathbb{N}} \left( \sum_{i \in L_t^+, i \in G} |y_i + \eta| + \sum_{i \in L_t^-, i \in G} |-y_i - \eta| \right)$$

$$\geq \sum_{t \in (\frac{\ln 1549}{\ln 1.001}, \frac{0.9999 \ln n}{\ln 1.001}) \cap \mathbb{N}} 0.499n \cdot \frac{1}{\pi} \frac{\ln 1.001}{1.001^t} \cdot 2 \cdot 1.001^t \cdot \beta$$

$$\geq \beta \cdot \frac{0.997}{\pi} n \ln n$$

The second inequality follows by Equation (10) and the triangle inequality, i.e., $\forall a, b, c \in \mathbb{R}, |a + c| + |b - c| \geq |a + b|$. $\qquad\square$

### B.7 Cost from the Sign-Disagreement Part of the Small-Entry Part

**Lemma B.17.** *Given a vector $y \in \mathbb{R}^n$ and a random vector $\Delta \in \mathbb{R}^n$ with $\Delta_i \sim C(0, 1)$ independently, with probability at least $1 - 1/2^{n^{\Theta(1)}}$,*

$$\sum_{i\,:\,\mathrm{sign}(y_i)\neq\mathrm{sign}(\Delta_i)\text{ and }|\Delta_i|<n^{0.9999}} |y_i + \Delta_i| > \frac{0.03}{\pi} n \ln n.$$

*Proof.* For $t \in [0, \frac{0.9999 \ln n}{\ln 4}) \cap \mathbb{N}$ define
$$L_t = \{i \in [n] \mid \mathrm{sign}(y_i) \neq \mathrm{sign}(\Delta_i), |\Delta_i| \in (4^t, 4^{t+1}], |\Delta_i| \notin [|y_i| - 4^t, |y_i| + 4^t]\}.$$
$\forall x \geq 1, y > 0$, we have

$$\Pr_{X \sim C(0,1)}[|X| \in (x, 4x], |X| \notin [y - x, y + x]]$$
$$\geq \Pr_{X \sim C(0,1)}[|X| \in (3x, 4x]]$$
$$= \frac{2}{\pi} \cdot (\tan^{-1}(4x) - \tan^{-1}(3x))$$
$$\geq \frac{0.1}{\pi} \cdot \frac{\ln(4)}{x}$$

Thus, $\forall i \in [n], t \in [0, \frac{0.9999 \ln n}{\ln 4}) \cap \mathbb{N}$,

$$\Pr[i \in L_t] \geq \frac{0.05}{\pi} \cdot \frac{\ln(4)}{4^t}.$$

Thus, $\forall t \in [0, \frac{0.9999 \ln n}{\ln 4}) \cap \mathbb{N}, \mathbf{E}[|L_t|] \geq 0.05n/\pi \cdot \ln(4)/4^t \geq n^{\Theta(1)}$. By a Chernoff bound and a union bound, with probability at least $1 - 1/2^{n^{\Theta(1)}} \forall t \in [0, \frac{0.9999 \ln n}{\ln 4}) \cap \mathbb{N}, |L_t| \geq 0.04n/\pi \cdot \ln(4)/4^t$. Thus, we have, with probability at least $1 - 1/2^{n^{\Theta(1)}}$,

$$\sum_{i \,:\, \text{sign}(y_i) \neq \text{sign}(\Delta_i) \text{ and } |\Delta_i| < n^{0.9999}} |y_i + \Delta_i|$$
$$\geq \sum_{t \in [0, \frac{0.9999 \ln n}{\ln 4}) \cap \mathbb{N}} |L_t| \cdot 4^t$$
$$\geq \frac{0.03}{\pi} n \ln n.$$

$\square$

## B.8  Overall Cost of the Small-Entry Part

**Lemma B.18** (For each). *Let $u = \eta \cdot \mathbf{1} \in \mathbb{R}^n$ where $\eta \in \mathbb{R}$ is an arbitrary real number. Let $\alpha \in \mathbb{R}^d$ where $\|\alpha\|_1 \geq 1 - 10^{-20}$. Let $\Delta \in \mathbb{R}^{n \times (d+1)}$ and $\forall (i,j) \in [n] \times [d+1], \Delta_{i,j} \sim C(0,1)$ are i.i.d. standard Cauchy random variables. Then with probability at least $1 - 1/2^{n^{\Theta(1)}}$,*

$$\sum_{j \in [n], |\Delta_{j,d+1}| < n^{0.9999}} |(u + \Delta_{d+1} - (u\mathbf{1}^\top + \Delta_{[d]})\alpha)_j| \geq \frac{2.025}{\pi} n \ln n.$$

*Proof.* Let $G_1 = \{j \in [n] \mid |\Delta_{j,d+1}| < n^{0.9999}, \text{sign}((u(1 - \mathbf{1}^\top \alpha) - \Delta_{[d]}\alpha)_j) = \text{sign}(\Delta_{d+1})_j)\}, G_2 = \{j \in [n] \mid |\Delta_{j,d+1}| < n^{0.9999}, \text{sign}((u(1 - \mathbf{1}^\top \alpha) - \Delta_{[d]}\alpha)_j) \neq \text{sign}(\Delta_{d+1})_j)\}$. Notice that $\Delta_{[d]}\alpha$ is a random vector with each entry independently drawn from $C(0, \|\alpha\|_1)$. Then with probability at least $1 - 1/2^{n^{\Theta(1)}}$,

$$\sum_{j \in [n], |\Delta_{j,d+1}| < n^{0.9999}} |(u + \Delta_{d+1} - (u\mathbf{1}^\top + \Delta_{[d]})\alpha)_j|$$
$$= \sum_{j \in [n], |\Delta_{j,d+1}| < n^{0.9999}} |(u(1 - \mathbf{1}^\top \alpha) - \Delta_{[d]}\alpha + \Delta_{d+1})_j|$$
$$= \sum_{j \in G_1} |(u(1 - \mathbf{1}^\top \alpha) - \Delta_{[d]}\alpha + \Delta_{d+1})_j| + \sum_{j \in G_2} |(u(1 - \mathbf{1}^\top \alpha) - \Delta_{[d]}\alpha + \Delta_{d+1})_j|$$
$$= \sum_{j \in G_1} |(u(1 - \mathbf{1}^\top \alpha) - \Delta_{[d]}\alpha)_j| + \sum_{j \in G_1} |(\Delta_{d+1})_j| + \sum_{j \in G_2} |(u(1 - \mathbf{1}^\top \alpha) - \Delta_{[d]}\alpha + \Delta_{d+1})_j|$$
$$\geq \|\alpha\|_1 \cdot \frac{0.997}{\pi} \cdot n \ln n + \frac{0.9998}{\pi} n \ln n + \frac{0.03}{\pi} n \ln n$$
$$\geq \frac{2.025}{\pi} n \ln n$$

The first inequality follows by Lemma B.16, Lemma B.14 and Lemma B.17. The second inequality follows by $\|\alpha\|_1 \geq 1 - 10^{-20}$. $\square$

**Lemma B.19** (For all). *Let $c > 0, c_0 > 0$ be two arbitrary constants. Let $u = \eta \cdot \mathbf{1} \in \mathbb{R}^n$ where $\eta \in \mathbb{R}$ satisfies $|\eta| \leq n^{c_0}$. Consider a random matrix $\Delta \in \mathbb{R}^{n \times (d+1)}$ with $d = n^{o(1)}$ and $\forall (i,j) \in [n] \times [d+1], \Delta_{i,j} \sim C(0,1)$ are i.i.d. standard Cauchy random variables. Conditioned on $\|\Delta\|_1 \leq n^3$, with probability at least $1 - 1/2^{n^{\Theta(1)}}, \forall \alpha \in \mathbb{R}^d$ with $1 - 10^{-20} \leq \|\alpha\|_1 \leq n^c$,*

$$\sum_{j \in [n], |\Delta_{j,d+1}| < n^{0.9999}} |(u + \Delta_{d+1} - (u\mathbf{1}^\top + \Delta_{[d]})\alpha)_j| \geq \frac{2.024}{\pi} n \ln n.$$

*Proof.* Let $\mathcal{N}$ be a set of points:

$$\mathcal{N} = \left\{ \alpha \in \mathbb{R}^d \mid 1 - 10^{-20} \leq \|\alpha\|_1 \leq n^c \text{ and } \exists q \in \mathbb{Z}^d, \text{ such that } \alpha = q/n^{c+c_0+1000} \right\}.$$

Since $d = n^{o(1)}$, we have $|\mathcal{N}| \leq (n^{2c+c_0+2000})^d = 2^{n^{o(1)}}$. By Lemma B.18 and a union bound, with probability at least $1 - 1/2^{n^{\Theta(1)}} \cdot |\mathcal{N}| \geq 1 - 1/2^{n^{\Theta(1)}}$, $\forall \alpha \in \mathcal{N}$, we have

$$\sum_{j \in [n], |\Delta_{j,d+1}| < n^{0.9999}} |(u + \Delta_{d+1} - (u\mathbf{1}^\top + \Delta_{[d]})\alpha)_j| \geq \frac{2.025}{\pi} n \ln n.$$

Due to the construction of $\mathcal{N}$, we have $\forall \alpha \in \mathbb{R}^d$ with $1 - 10^{-20} \leq \|\alpha\|_1 \leq n^c$, $\exists \alpha' \in \mathcal{N}$ such that $\|\alpha - \alpha'\|_\infty \leq 1/n^{c+c_0+1000}$. Let $\gamma = \alpha - \alpha'$. Then

$$\sum_{j \in [n], |\Delta_{j,d+1}| < n^{0.9999}} |(u + \Delta_{d+1} - (u\mathbf{1}^\top + \Delta_{[d]})\alpha)_j|$$

$$= \sum_{j \in [n], |\Delta_{j,d+1}| < n^{0.9999}} |(u + \Delta_{d+1} - (u\mathbf{1}^\top + \Delta_{[d]})(\alpha' + \gamma))_j|$$

$$\geq \sum_{j \in [n], |\Delta_{j,d+1}| < n^{0.9999}} |(u + \Delta_{d+1} - (u\mathbf{1}^\top + \Delta_{[d]})\alpha')_j| - \sum_{j \in [n], |\Delta_{j,d+1}| < n^{0.9999}} |((u\mathbf{1}^\top + \Delta_{[d]})\gamma)_j|$$

$$\geq \sum_{j \in [n], |\Delta_{j,d+1}| < n^{0.9999}} |(u + \Delta_{d+1} - (u\mathbf{1}^\top + \Delta_{[d]})\alpha')_j| - \|(u\mathbf{1}^\top + \Delta_{[d]})\gamma\|_1$$

$$\geq \frac{2.025}{\pi} n \ln n - 1/n^{500}$$

$$\geq \frac{2.024}{\pi} n \ln n$$

The first equality follows from $\alpha = \alpha' + \gamma$. The first inequality follows by the triangle inequality. The third inequality follows from $\|\gamma\|_1 \leq 1/n^{c+c_0+800}$, $\|u\mathbf{1}^\top\|_1 \leq n^{c_0+10}$, $\|\Delta\|_1 \leq n^3$, and $\forall \alpha' \in \mathcal{N}$,

$$\sum_{j \in [n], |\Delta_{j,d+1}| < n^{0.9999}} |(u + \Delta_{d+1} - (u\mathbf{1}^\top + \Delta_{[d]})\alpha')_j| \geq \frac{2.025}{\pi} n \ln n.$$

$\square$

## B.9 Main result

**Theorem B.20** (Formal version of Theorem 1.2)**.** *Let $n > 0$ be sufficiently large. Let $A = \eta \cdot \mathbf{1} \cdot \mathbf{1}^\top + \Delta \in \mathbb{R}^{n \times n}$ be a random matrix where $\eta = n^{c_0}$ for some sufficiently large constant $c_0$, and $\forall i, j \in [n], \Delta_{i,j} \sim C(0,1)$ are i.i.d. standard Cauchy random variables. Let $r = n^{o(1)}$. Then with probability at least $1 - O(1/\log\log n)$, $\forall S \subset [n]$ with $|S| = r$,*

$$\min_{X \in \mathbb{R}^{r \times n}} \|A_S X - A\|_1 \geq 1.002 \|\Delta\|_1$$

*Proof.* We first argue that for a fixed set $S$, conditioned on $\|\Delta\|_1 \leq 100 n^2 \ln n$, with probability at least $1 - 1/2^{n^{\Theta(1)}}$,

$$\min_{X \in \mathbb{R}^{r \times n}} \|A_S X - A\|_1 \geq 1.002 \|\Delta\|_1.$$

Then we can take a union bound over the at most $n^r = 2^{n^{o(1)}}$ possible choices of $S$. It suffices to show for a fixed set $S$, $\min_{X \in \mathbb{R}^{r \times n}} \|A_S X - A\|_1$ is not small.

Without loss of generality, let $S = [r]$, and we want to argue the cost

$$\min_{X \in \mathbb{R}^{r \times n}} \|A_S X - A\|_1 \geq \min_{X \in \mathbb{R}^{r \times n}} \|A_S X_{[n] \setminus S} - A_{[n] \setminus S}\|_1 \geq 1.002 \|\Delta\|_1.$$

Due to Lemma B.2, with probability at least $1 - O(1/\log\log n)$, $\|\Delta\|_1 \leq 4.0002/\pi \cdot n^2 \ln n$. Now, we can condition on $\|\Delta\|_1 \leq 4.0002/\pi \cdot n^2 \ln n$.

Consider $j \in [n] \setminus S$. Due to Lemma B.6, with probability at least $1 - (1/n)^{\Theta(n)}$, for all $X_j \in \mathbb{R}^r$ with $\|X_j\|_1 \geq n^c$ for some constant $c > 0$, we have

$$\|A_S X_j - A_j\|_1 = \|(\eta \cdot \mathbf{1} \cdot \mathbf{1}^\top + [\Delta_S \ \Delta_j])[X_j^\top \ -1]^\top\|_1 \geq 0.9n^3.$$

By taking a union bound over all $j \in [n] \setminus S$, with probability at least $1 - (1/n)^{\Theta(n)}$, for all $X \in \mathbb{R}^{r \times n}$ with $\exists j \in [n] \setminus S, \|X_j\|_1 \geq n^c$, we have

$$\|A_S X - A\|_1 \geq 0.9n^3.$$

Thus, we only need to consider the case $\forall j \in [n] \setminus S, \|X_j\|_1 \leq n^c$. Notice that we condition on $\|\Delta\|_1 \leq 4.0002/\pi \cdot n^2 \ln n$. By Fact B.1, we have that if $\|X_j\|_1 \leq n^c$ and $|1 - \mathbf{1}^\top X_j| > 1 - 10^{-20}$, then $\|A_S X - A\|_1 \geq \|A_S X_j - A_j\|_1 > n^3$.

Thus, we only need to consider the case $\forall j \in [n] \setminus S, \|X_j\|_1 \leq n^c, |1 - \mathbf{1}^\top X_j| \leq 1 - 10^{-20}$. $\forall X \in \mathbb{R}^{r \times n}$ with $\forall j \in [n] \setminus S, \|X_j\|_1 \leq n^c, |1 - \mathbf{1}^\top X_j| \leq 1 - 10^{-20}$, if $\|A_S X_{[n] \setminus S} - A_{[n] \setminus S}\|_1 \leq 4n^2 \ln n$, then

$$
\begin{aligned}
&\|A_S X_{[n] \setminus S} - A_{[n] \setminus S}\|_1 \\
&= \|(\eta \cdot \mathbf{1} \cdot \mathbf{1}^\top + \Delta_S)X_{[n] \setminus S} - (\eta \cdot \mathbf{1} \cdot \mathbf{1}^\top + \Delta_{[n] \setminus S})\|_1 \\
&\geq \sum_{i \in [n], j \in [n] \setminus S, |\Delta_{i,j}| \geq n^{1.0002}} |(((\eta \cdot \mathbf{1} \cdot \mathbf{1}^\top + \Delta_S)X_{[n] \setminus S} - \eta \cdot \mathbf{1} \cdot \mathbf{1}^\top) - \Delta_{[n] \setminus S})_{i,j}| \\
&\quad + \sum_{i \in [n], j \in [n] \setminus S, |\Delta_{i,j}| < n^{0.9999}} |((\eta \cdot \mathbf{1} \cdot \mathbf{1}^\top + \Delta_S)X_{[n] \setminus S} - (\eta \cdot \mathbf{1} \cdot \mathbf{1}^\top + \Delta_{[n] \setminus S}))_{i,j}| \\
&\geq \frac{1.996}{\pi} \cdot n^2 \ln n + \sum_{i \in [n], j \in [n] \setminus S, |\Delta_{i,j}| < n^{0.9999}} |((\eta \cdot \mathbf{1} \cdot \mathbf{1}^\top + \Delta_S)X_{[n] \setminus S} - (\eta \cdot \mathbf{1} \cdot \mathbf{1}^\top + \Delta_{[n] \setminus S}))_{i,j}| \\
&= \frac{1.996}{\pi} \cdot n^2 \ln n + \sum_{j \in [n] \setminus S} \sum_{i \in [n], |\Delta_{i,j}| < n^{0.9999}} |((\eta \cdot \mathbf{1} \cdot \mathbf{1}^\top + \Delta_S)X_j - \eta \cdot \mathbf{1} - \Delta_j)_i| \\
&\geq \frac{1.996}{\pi} \cdot n^2 \ln n + \sum_{j \in [n] \setminus S} \frac{2.024}{\pi} n \ln n \\
&\geq \frac{1.996}{\pi} \cdot n^2 \ln n + \frac{2.023}{\pi} n^2 \ln n \\
&\geq \frac{4.01}{\pi} \cdot n^2 \ln n
\end{aligned}
$$

holds with probability at least $1 - 1/2^{n^{\Theta(1)}}$. The first equality follows by the definition of $A$. The first inequality follows by the partition by $|\Delta_{i,j}|$. Notice that $[\mathbf{1} \ \Delta_S]$ has rank at most $r + 1 = n^{o(1)}$. Then, due to Lemma B.13, and the condition $\|A_S X_{[n] \setminus S} - A_{[n] \setminus S}\|_1 \leq 4n^2 \ln n$, the second inequality holds with probability at least $1 - 1/2^{n^{\Theta(1)}}$. The second equality follows by grouping the cost by each column. The third inequality holds with probability at least $1 - 1/2^{n^{\Theta(1)}}$ by Lemma B.19, and a union bound over all the columns in $[n] \setminus S$. The fourth inequality follows by $n - r = n - n^{o(1)} \geq (1 - 10^{-100})n$.

Thus, conditioned on $\|\Delta\|_1 \leq 4.0002/\pi \cdot n^2 \ln n$, with probability at least $1 - 1/2^{n^{\Theta(1)}}$, we have $\min_{X \in \mathbb{R}^{r \times n}} \|A_S X - A\|_1 \geq \frac{4.02}{\pi} \cdot n^2 \ln n$. By taking a union bound over all the $\binom{n}{r} = 2^{n^{o(1)}}$ choices of $S$, we have that conditioned on $\|\Delta\|_1 \leq \frac{4.0002}{\pi} n^2 \ln n$, with probability at least $1 - 1/2^{n^{\Theta(1)}}$, $\forall S \subset [n]$ with $|S| = r = n^{o(1)}$, $\min_{X \in \mathbb{R}^{r \times n}} \|A_S X - A\|_1 \geq \frac{4.02}{\pi} \cdot n^2 \ln n$. Since $4.01/4.0002 > 1.002$,

$$\min_{X \in \mathbb{R}^{r \times n}} \|A_S X - A\|_1 \geq 1.002\|\Delta\|_1.$$

Since $\|\Delta\|_1 \leq \frac{4.0002}{\pi} n^2 \ln n$ happens with probability at least $1 - O(1/\log\log n)$, this completes the proof. $\qquad \square$