[Reviews · NeurIPS 2019]

Reviewer 1



This paper studies low rank matrix approximation with respect to the entry-wise L1 norm. The main result is a theoretical guarantee for the low rank setting where an arbitrary rank k matrix is additionally perturbed by a random matrix. The result shows how to obtain a O(k\log n + poly(k/\epsilon)) rank matrix with error less than that of the L1 norm of the random matrix. The random matrix is subject to a certain moment condition which is shown to be necessary as well. The paper further provides a heuristic method inspired by the theoretical machinery. Strength: *) Strong upper bound which improves understanding of column subset selection for L1 loss, along with hardness results on assumptions needed in the upper bound *) The proposed method is conceptually simple, and indeed the paper evaluated a heuristic based on the proposed method Weakness: *) The empirical result only gives incremental improvement over prior methods on the two real datasets, especially for larger values of k (rank). *) Discrepancy between the algorithm analyzed and the evaluated More detailed questions: *) Line 64-65, what is the order of the polynomial? Wouldn't a high polynomial dependency on \epsilon be less desirable as well? *) How well do Algorithm 2 apply in the setting of equation (2) (i.e. when you are trying to compute the best rank-k approximation)? Could the same idea be applied to get improvement in that setting? *) Algorithm 1 line 6: S -> Q? Other remarks: the formatting of equations could be made better in the current format.

Reviewer 2



Originality: The results obtained in this work are, to the best of my knowledge, new. The methods followed are relatively new (especially the ones using the notions of tuples" and "cores" to tackle the distributional assumptions on the noise. ) Quality: Both the statements of the results and the proofs I checked appear technically sound and correct. I am though skeptical about the experiments section: the authors discuss and analyze Algorithm 1 for 7 pages and suddenly they implement Algorithm 2 which is an elementary median heuristic. It is interesting that it beats more sophisticated methods, but I am unfortunately not convinced the Algorithm 2 is linked with the averaging argument in the analysis of Algorithm 1, or that it is sufficiently connected with the rest of the paper. I think the authors should address this issue. Clarity: For the most part the paper is well written and the ideas clearly expressed. I think though some parts of the paper are not up the same standard and are slightly poorly written: e.g. parts of subsections 1.2. and 2.2. seems a little bit rushed; for 1.2. the authors miss to explain at the end of page 3 what is the cardinality of the random sets I_t and the number of them (an important step) and for subsection 2.2. the argument using Cramer Rule which is hiding behind the relatively complex definitions should be clearly explained as it can help significantly the reader. Significance: I think the methods followed are significant, but I do not consider clear their significance beyond the setting considered. For example, is the averaging considered tied to \ell_1 loss or can it generalize to the \ell_p norm loss?

Reviewer 3



This paper introduces an algorithm to obtain a low-rank reconstruction of a nxn matrix under the L_1 loss, with provable guarantees under mild assumptions on the distribution of the noise. My main question for the authors relates to their parallel submission #3303, which was flagged as a potential dual submission (allowing reviewers to ask questions as if the other work were already published). My reading of both papers does *not* suggest that these works are too closely related to be both accepted. However, I would appreciate the authors providing a comparison of the results of both contributions. - Assuming the necessary conditions for both papers are met, Thm. 1.2 of #3303 for the L1 loss states that we can in O(n^2 + n poly(d log n)) time find a subset S of size O(k log n) s.t. ||A_S X - A||_1 <= O(k log k) ||Delta||_1 w.p. 0.99. (My apologies for the inline math). Am I understanding correctly that the contribution of this paper allows a tradeoff in the computational time to obtain a better approximation by way of the epsilon parameter? - More generally, how do the bounds you derive in this paper compare to the ones that would be obtained when using the L1 loss in #3303 (with all necessary assumptions)? Are there specific scenarios where the bounds from either paper should be preferred? - You mention in Section 3 that taking the median performs better in practice than taking the average. It might be worth showing this empirically. I would also be curious to know how the compared algorithms performed on average and the order of magnitude of the standard deviation for all reconstruction methods, as the authors only provide a comparison between the best results of each reconstruction algorithm. Minor comments: - I believe the left hand side of Eq. (3) of the appendix should read \int_0^n rather than ^\infty. ******** Post rebuttal comments ******** I am satisfied with the author's response regarding the use of mean vs. average. However, if the paper is accepted, I ask the reviewers to include their experimental results for the average, as it is the method suggested by their theoretical analysis.

Reviewer 4



This paper studies low rank approximation of the matrix A + Delta, where Delta consists of i.i.d. noise variables. The paper considers the L_1 cost function and makes only a mild assumption on the moment of the noises. The new algorithm proposed by the author is a column-selection type algorithm. Roughly speaking, the authors first argue that there is only a small subset of columns that have large entries (coming from the tail of the noises). An existing algorithm can be used to approximate these columns. For the rest of small entries, the author argues that using probabilistic method to randomly choose columns suffices. I think this is a fairly strong theoretical result. While the general framework may not be entirely new (i.e., identify heavy hitters and use a different strategy to deal with heavy hitters), there are quite a few interesting technical innovations, e.g., building specialized concentration bounds optimized for their own analysis. I start to be unable to follow technical details after Sec 2.2 (see below for more questions). My major concern is that the success probability is only a constant (0.99). It is not clear to me whether the success probability is with respect to the data or with respect to the random tosses in the algorithm (i.e., whether the success probability can be boosted with more running time). If the success probability is with respect to the data, I would consider it a much weaker result. A "standard requirement" for the failure probability is $1/n^c$ for a sufficiently large constant $c$. Technically, it may not even circumvent the lower bound from [24] as advertised by the authors --- information theoretic lower bounds usually are in the form "any algorithm fails with constant probability" and this is consistent with such kind of result. The most worrying building block is Lemma 2.3 (where a constant failure probability with respect to data shows up). There, only a Markov inequality is used so it seems unlikely to significantly push down the failure probability. This is also consistent with properties of "heavy tail" noises --- proving tail bounds for such noises is usually difficult. Detailed questions: 1. I am not able to understand Definition R_{A^*}(S) (line 220). Specifically, Q is not explained in the definition. 2. Averaging reduces noises (around Lemma 2.2). It looks this is an important technique but I am unable to see the high level intuition. The paper spent considerable effort to explain a simple fact resembling central limit theorem/law of large number but then it was hand-waiving in explaining how this simple fact may be used.

[Author Response · NeurIPS 2019]

**Response to Reviewer #1**

[Discrepancy between the algorithm analyzed and the evaluated]: Our theory motivated the algorithm we implemented, though we tweaked it empirically and observed improvements by taking the median. Indeed, our theory shows that aggregating multiple columns helps reduce the cost for the $\ell_1$-loss in our distributional setting, which is known to be impossible in a worst-case setting (see below). We initially implemented the average, and that performed well as predicted by our theory, but taking the median performed even better on our datasets. However we agree that the latter does not have theoretical guarantees, and so we are also happy and will include figures for taking the average.

[Order of the polynomial in line 64-65]: The exponent depends on the exponent of the polynomial of the $\ell_1$-regression solver used. If $\ell_1$-regression solvers improve, then the running time of our algorithm also improves. Furthermore, the order of the polynomial also depends on the constant $p$. For larger $p$, the order of the polynomial is smaller. We did not explicitly compute the order, but we stress that before our work no polynomial of any order was known.

[Apply Algorithm 2 for best rank-$k$ approximation]: We do not know if the median algorithm can be analyzed for general $k$ to give a theoretical result for low rank approximation with the $\ell_1$-loss, and leave this as an intriguing open question, inspired by our experiments.

**Response to Reviewer #2**

[Median heuristic]: See response to Reviewer #1 above.

[Clarity of Subsection 1.2 and Subsection 2.2]: At the end of page 3, we should mention that the cardinality of a random set is at most $\text{poly}(k/\varepsilon)$ and there are $\text{poly}(k/\varepsilon)$ of them. In Subsection 2.2, the use of Cramer's rule is as follows. Consider a rank $k$ matrix $M \in \mathbb{R}^{n \times (k+1)}$. Let $P \subseteq [k+1], Q \subseteq [n], |P| = |Q| = k$ be such that $|\det(M_P^Q)|$ is maximized. Since $M$ has rank $k$, we know $\det(M_P^Q) \neq 0$ and thus the columns of $M_P$ are independent. Let $i \in [k+1] \setminus P$. Then the linear equation $M_P x = M_i$ is feasible and there is a unique solution $x$. Furthermore, by Cramer's rule $x_j = \det(M_{[k+1]\setminus\{j\}}^Q)/\det(M_P^Q)$. Since $|\det(M_P^Q)| \geq |\det(M_{[k+1]\setminus\{j\}}^Q)|$, we have $\|x\|_\infty \leq 1$.

[Applicability/significance of the method beyond the $\ell_1$-loss]: Our averaging idea is very general and we expect it to be useful for other loss functions, which we leave as an open question. We focused on $\ell_1$ since there is a known lower bound for column subset selection with $\ell_1$-loss (see [24], discussed more below), which we bypass in our setting.

**Response to Reviewer #3**

[Overall comparison and comparing with the bounds obtained when $\ell_1$-loss is used in #3303]: The other paper concerns characterizing when one can obtain good low rank approximations for arbitrary loss functions and is for worst-case inputs, while for the $\ell_1$-loss it is known [24] that it is impossible to obtain subsets of fewer than $\text{poly}(k)$ columns spanning any $\sqrt{k}$-approximate low rank approximation. Here, under our distributional assumptions we are able to get very accurate, $(1+\varepsilon)$-approximate low rank approximations for $\ell_1$ with only $O(k \log n) + \text{poly}(k/\varepsilon)$ columns. Further, we show our distributional assumption is necessary. Thus, the other paper #3303 cannot be used to obtain a $(1+\varepsilon)$-approximation since it is simply not possible without distributional assumptions – indeed, as the reviewer states, that paper obtains an $O(k \log k)$ approximation. Further, showing that our distributional assumptions suffice is quite involved. We believe when high accuracy is required, this paper should be used, while when one wants an arbitrary loss function, which might not be scale-invariant (so well beyond $\ell_p$ norms, e.g., Huber loss), then one should use that paper.

[When necessary conditions are met for both papers]:The tradeoff is not only between time and approximation but also between the number of columns selected and the approximation. #3303 selects $O(k \log n)$ columns but only achieves $O(k \log k)$ approximation while this paper selects $O(k \log n) + \text{poly}(k/\varepsilon)$ columns and achieves $(1+\varepsilon)$ approximation. Technically, this paper mainly focuses on showing how an extra $\text{poly}(k/\varepsilon)$-sized sample of columns can help reduce the error to $(1+\varepsilon)$ while #3303 focuses on why a recursive filtering approach works for general loss functions.

[Mean and standard deviation]: We repeated each experiment 25 times. In all experiment settings, our average approximation ratio is the best among compared algorithms. In addition, the standard deviation is also small. For example, for mfeat dataset, $k = 1$ and 1.1-stable random noise, the mean approximation ratio of SVD, L1LowRank, Uniform and Ours are $3.05, 4.12, 1.27, 1.01$ respectively, and the standard deviations are $0.47, 1.46, 0.09, 0.001$ respectively. The randomness is over both the data and the algorithm. Due to the page limit of the response, we will report all means and standard deviations in the final version.

**Response to Reviewer #4**

[Success probability]: Though the success probability is with respect to both the data and the randomness of the algorithm, all stated .999 probabilities can be made $1 - n^{-\Omega(1)}$. In Lemma 2.3, the size of $H$ (the set of columns with large entries) slightly changes, and this changes line 297, where $T' \leq O(\cdots + |H|)$. Notice that in line 290, $qt$ is $n^{o(1)}$, so no change is needed. In Lemma 2.3, we can make $|H| \leq n^{1-(p-1)/2+\delta}$ for some sufficiently small positive constant $\delta \leq (p-1)/2$ (e.g., $\delta = (p-1)/4$), and still use Markov's inequality. This is desired since what we want is to make $|H| \leq n^{1-\Omega(1)}$. Furthermore, this will happen with probability at least $1 - n^{-\delta} = 1 - n^{-\Omega(1)}$.

[Detailed question]: 1. The max is over $P$ and $Q$ while $R_{A^*}(S)$ only takes the value of the corresponding $P$. 2. The reason that we spent considerable effort is that our moment condition seems to be a considerably weaker assumption than typical assumptions, e.g., it is much weaker than assuming a bounded variance, and thus while it resembles central limit theorems, it is definitely different as the random variables we consider may not even have a variance.

[Meta-Review · NeurIPS 2019]

The reviewers are all in agreement that the paper makes significant enough contributions to merit acceptance. The authors should take the reviewer comments, particularly those that required elaborate feedback, into account when preparing the final version.